# Animal Models of Drug-Resistant Epilepsy as Tools for Deciphering the Cellular and Molecular Mechanisms of Pharmacoresistance and Discovering More Effective Treatments

**DOI:** 10.3390/cells12091233

**Published:** 2023-04-24

**Authors:** Wolfgang Löscher, H. Steve White

**Affiliations:** 1Department of Pharmacology, Toxicology, and Pharmacy, University of Veterinary Medicine, Bünteweg 17, 30559 Hannover, Germany; 2Center for Systems Neuroscience, 30559 Hannover, Germany; 3Department of Pharmacy, School of Pharmacy, University of Washington, Seattle, WA 98195, USA

**Keywords:** seizures, anti-seizure drugs, drug resistant epilepsy, kindling, pilocarpine, kainate, Dravet syndrome, temporal lobe epilepsy

## Abstract

In the last 30 years, over 20 new anti-seizure medicines (ASMs) have been introduced into the market for the treatment of epilepsy using well-established preclinical seizure and epilepsy models. Despite this success, approximately 20–30% of patients with epilepsy have drug-resistant epilepsy (DRE). The current approach to ASM discovery for DRE relies largely on drug testing in various preclinical model systems that display varying degrees of ASM drug resistance. In recent years, attempts have been made to include more etiologically relevant models in the preclinical evaluation of a new investigational drug. Such models have played an important role in advancing a greater understanding of DRE at a mechanistic level and for hypothesis testing as new experimental evidence becomes available. This review provides a critical discussion of the pharmacology of models of adult focal epilepsy that allow for the selection of ASM responders and nonresponders and those models that display a pharmacoresistance per se to two or more ASMs. In addition, the pharmacology of animal models of major genetic epilepsies is discussed. Importantly, in addition to testing chemical compounds, several of the models discussed here can be used to evaluate other potential therapies for epilepsy such as neurostimulation, dietary treatments, gene therapy, or cell transplantation. This review also discusses the challenges associated with identifying novel therapies in the absence of a greater understanding of the mechanisms that contribute to DRE. Finally, this review discusses the lessons learned from the profile of the recently approved highly efficacious and broad-spectrum ASM cenobamate.

## 1. Introduction

Resistance to anti-seizure medications (ASMs) remains one of the major problems in the treatment of epilepsy, affecting about one-third of patients with seizures [1,2]. Animal models of drug-resistant epilepsy (DRE) can provide an important tool to help understand the cellular and molecular basis underlying resistance to ASMs and to develop new, more effective treatments [2,3,4,5,6]. According to Kwan et al. [7], DRE is defined as the failure of adequate trials of two tolerated, appropriately chosen, and used ASM schedules (whether as monotherapies or in combination) to achieve sustained seizure freedom, so that animal models of DRE should meet this definition [8]. The probability of intractability largely depends on the type of seizures and epilepsy, with focal seizures such as those occurring in temporal lobe epilepsy (TLE) having the poorest prognosis of all seizure types in adults [9,10]. This explains why almost all animal models of DRE are models of TLE. However, drug resistance is also a major problem in childhood epilepsies such as the Dravet and Lennox–Gastaut syndromes and tuberous sclerosis complex [11]. Animal models of such difficult-to-treat childhood epilepsies are increasingly being used for pharmacological studies (see Section 4.3), but the present review will be mainly restricted to models of TLE.

DRE is a major health problem, associated with increased morbidity and mortality and accounting for much of the economic burden of epilepsy [9,12]. The problem of intractable or difficult-to-treat seizures has not been changed to any significant extent by the introduction of various new ASMs, although drug treatment has become better tolerable for some patients [13,14]. A major obstacle in developing new strategies for the treatment of DRE is that mechanisms of refractoriness are only poorly understood. Some clinical features are associated with pharmacoresistance, including early onset of seizures (before one year of age), high seizure frequency (or density) before the onset of treatment, a history of febrile seizures, the type of seizures (about 60% of patients with intractable epilepsy suffer from focal seizures) or epilepsy, structural brain lesions, malformations of cortical development, and psychiatric comorbidities [9,10,15]. However, relatively little research has been undertaken to understand the basis of these associations.

There are many possible causes of DRE: it is likely to be a multifactorial process [2]. Genetic factors, e.g., gene polymorphisms, may be important and explain why two patients with the same type of epilepsy or seizures may differ in their response to ASMs. Disease-related factors are certainly important, including the etiology of the seizures, progression of epilepsy under treatment with ASMs, alterations in drug targets, or alterations in drug uptake into the brain. Furthermore, drug-related factors also likely contribute to insufficient seizure control, including loss of anti-seizure efficacy during treatment, i.e., development of tolerance, or ineffective mechanisms of action of currently available ASMs in patients with medically intractable epilepsy.

An important characteristic of DRE is that most patients with DRE are resistant to several mechanistically distinct ASMs [9,10,14,16]. As a consequence, patients not controlled on monotherapy with the first ASM have a chance of only about 10–15% to be controlled by other ASMs, even when using ASMs that act by diverse mechanisms. This argues against epilepsy-induced alterations in specific drug targets as a major cause of pharmacoresistant epilepsy and points to nonspecific and possibly adaptive mechanisms, such as decreased drug uptake into epileptic brain regions by local overexpression of multidrug transporters at the blood–brain barrier (BBB) or loss of drug targets in epileptic brain regions by neuronal damage [2].

Because of the problem of intractable epilepsy, there are increasingly more animal models specifically dedicated to identifying effective therapeutic agents for resistant epilepsy or to studying mechanisms of drug resistance [3,5,6,17,18]. An animal model of epilepsy allowing the selection of subgroups of animals with drug-resistant and drug-responsive seizures could be a valuable tool to study why and how seizures become intractable and to develop more effective treatment strategies. Löscher became interested in developing such an animal model some 35 years ago [19], leading to the discovery of phenytoin-resistant subgroups of amygdala-kindled Wistar rats [20]. Subsequently, Löscher’s group also studied whether rats with spontaneous recurrent seizures (SRSs) developing after status epilepticus (SE) differ in their individual sensitivity to ASMs [21,22,23,24,25]. Another strategy of developing animal models of drug-resistant seizures was chosen by Steve White’s group, leading to the 6 Hz corneal stimulation seizure model, originally described as a model of “psychomotor seizures” by Toman [26,27] and later proposed as a model of drug-resistant focal epilepsy [6], and the lamotrigine-resistant kindled rat [28]. Furthermore, the intrahippocampal kainate (IHK) model in mice has become a widely used model of drug-resistant SRSs [29].

Concerning the models described and discussed in this review, it is important to consider that the term “drug-resistant” (or pharmacoresistant or drug refractory) applied in the context of animal models, based on experience in patients with epilepsy, can be minimally defined as persistent seizure activity not responding or with very poor response to monotherapy with at least two current ASMs at the maximum tolerated doses [8]. A goal should be to develop animal models of DRE that reflect the most common types of intractable or difficult-to-treat epilepsy in humans. As mentioned, almost all of the currently available models are models of TLE.

The main goal of this review is to describe the use of animal models of drug-resistant seizures/epilepsy as tools for deciphering the cellular and molecular mechanisms of pharmacoresistance. A second goal is to illustrate their usefulness for discovering novel treatments that may be more effective than currently used ASMs. An overview of the animal models discussed here is shown in Figure 1.

## 2. Chronic Animal Models That Allow Selecting Drug Responders and Nonresponders

It is not well understood why two patients with the same type of epilepsy and seemingly identical seizure types may differ strikingly in their response to the same ASM, one becoming seizure-free and the other not, despite adequate drug choice and dosing. Therefore, starting in the 1980s, the strategy of Löscher’s group in animal model development has been to identify models in which it is possible to select ASM responders and nonresponders from the same group of animals, thus allowing to determine—by direct subgroup comparison—which mechanisms underlie the ASM-resistance in the nonresponders [18,30,31]. This strategy led to the development of three models, one with kindled seizures and two with SRSs, which will be described below.

### 2.1. The Amygdala Kindling Model of Temporal Lobe Epilepsy

Kindling refers to the phenomenon that animals chronically implanted with a stimulation electrode in one structure of the limbic system or other brain areas (the amygdala being among the most responsive structures) develop focal and secondarily generalized seizures of increasing severity and duration upon periodic (e.g., one daily) electrical stimulation with a short (e.g., 1 s) initially subconvulsive current. After its introduction in 1969 by Goddard et al. [32], kindling became one of the most widely used animal models of epilepsy, particularly because several of the mechanisms involved in kindling were thought to be relevant for the development of TLE, the most common and difficult-to-treat type of epilepsy in adults [3,33,34]. However, a major drawback of this model is that kindled rats, unless “over-kindled” (see Section 2.3.4), do not exhibit SRSs, so elicited seizures have to be used for drug studies. On the other hand, amygdala kindling belongs to the few models that have been validated clinically because the decision to develop the ASM levetiracetam for the clinical treatment of focal seizures was mainly based on its high efficacy in the amygdala kindling model [35,36,37], which correctly predicted anti-seizure efficacy against difficult-to-treat focal seizures in patients [3,37,38]. In addition to kindling, only the maximal electroshock seizure (MES) and the pentylenetetrazole (PTZ) seizure tests have been validated to predict the anti-seizure efficacy of novel ASMs in patients [3]; however, the latter acute seizure models in otherwise healthy rodents are not considered animal models of drug-resistant seizures but are mainly used for drug screening [39].

### 2.2. Amygdala Kindling as a Model of Pharmacoresistant Seizures

Löscher first proposed kindling as a model to investigate intractable epilepsy in 1986 [13,40]. Indeed, to the best of our knowledge, this was the first animal model of DRE that was proposed for this purpose in the literature. In all studies, Löscher’s group used the original protocol described by Goddard et al. (1969) with once daily electrical stimulation of the basolateral amygdala (BLA) until all rats are “fully kindled”, i.e., exhibit the same maximum response (a secondarily generalized stage five seizure) upon stimulation. By directly comparing standard ASMs in the kindling model and the standard MES test in age-matched female Wistar rats, Löscher found that kindled seizures were less sensitive to anti-seizure treatment than primarily generalized seizures as produced in the MES test (Figure 2 and Appendix A). Furthermore, in the kindling model, focal seizure stages were found to be much less responsive to ASMs than secondarily generalized convulsive seizures, which is consistent with clinical experience. Figure 2B also illustrates TD_50_s determined in the rotarod test (see Section 3 for more details), demonstrating that phenobarbital and valproate suppressed focal seizures in kindled rats only at doses that induced motor impairment in nonkindled rats. In this respect, it is important to note that kindled rats are more susceptible to “neurotoxic” adverse effects of several ASMs (e.g., carbamazepine, phenobarbital, valproate, and diazepam) than nonkindled rats [41], which is important when calculating “protective indices” for ASMs in kindled rats (see Section 3). Based on these data, Löscher proposed that the search for novel compounds with high potency in the amygdala-kindling model may be a promising strategy in the development of new ASMs for patients with intractable epilepsy [19,40,42].

**Figure 2 cells-12-01233-f002:**
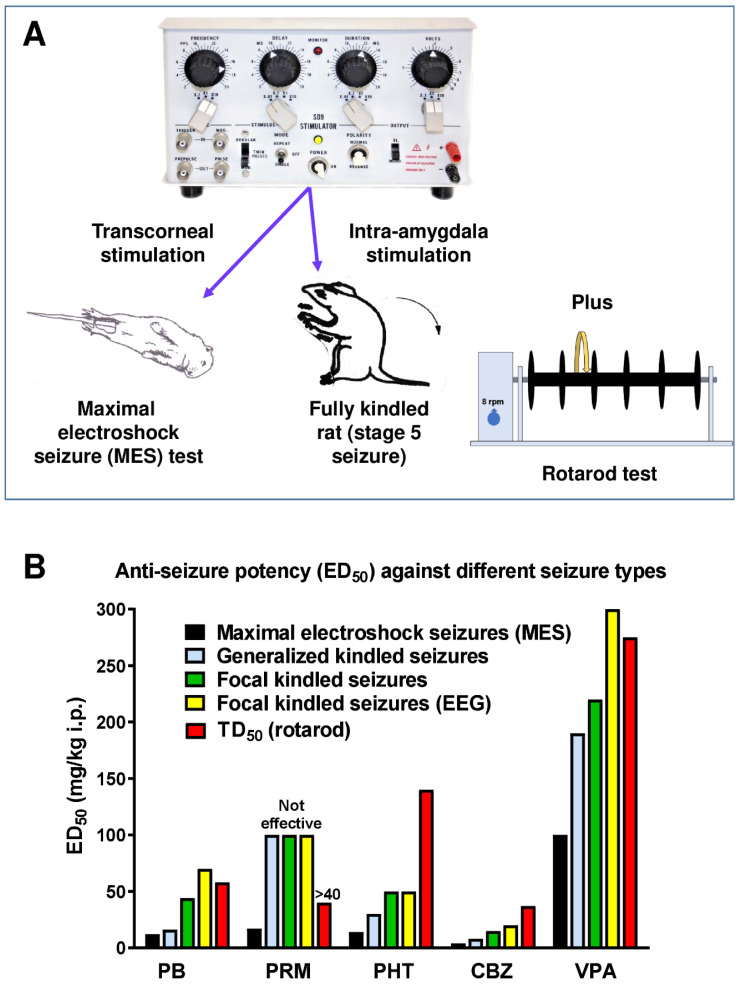
A comparison of the pharmacology of maximal electroshock seizures (MES) and different types of amygdala-kindled seizures in age-matched female Wistar rats. For comparison, also the TD_50_ determined in the rotarod test in nonkindled female Wistar rats is shown. However, note that the TD_50_ of ASMs may be significantly lower in kindled rats [41]. (**A**) Schematic illustration of the three models used for this comparison. (**B**) Anti-seizure potencies, expressed as i.p. ED_50_s and i.p. TD_50_s for “minimal neurotoxicity”. As illustrated, all kindled seizure types were more difficult to suppress than MES in naive (non-kindled) rats. Furthermore, in kindled rats, focal kindled seizures (stages 1–3) were less sensitive to ASMs than secondarily generalized kindled (stage 4–5) seizures. Similar results were found with benzodiazepines (diazepam, clonazepam; see Appendix A) and some investigational drugs, which are not illustrated in the figure. “Not effective” indicates that an ED_50_ could not be determined up to the indicated dose. Data are from Löscher et al. [40] and Löscher and Nolting [43]. Abbreviations: CBZ, carbamazepine; PB, phenobarbital; PHT, phenytoin; PRM, primidone; and VPA, valproate.

### 2.3. Selection of Phenytoin Responders and Nonresponders from Large Groups of Amygdala-Kindled Rats

In a subsequent study on the acute effects of phenytoin in amygdala-kindled rats, Löscher’s group found that, within a group of rats with fixed suprathreshold current stimulation, phenytoin was capable of totally blocking focal and generalized seizures in some of the animals, whereas in other rats phenytoin did not exert anti-seizure effects or even increased the duration of seizures, thus indicating marked variation in the individual response to this drug [44]. Rundfeldt et al. [44] suggested that this variation could be related to marked interindividual differences in phenytoin’s potency to increase the focal seizure threshold. Based on these findings, we proposed that kindled rats with a low response to phenytoin could be an ideal model for drug-resistant focal epilepsy.

This idea was tested prospectively by kindling a large group of animals and selecting animals with different responsiveness to phenytoin by determining the effect of phenytoin on the afterdischarge threshold (ADT), i.e., the threshold for induction of focal seizure activity via an amygdala electrode in kindled rats (Figure 3A). Phenytoin was repeatedly tested (3–4 times) at intervals of at least 5 days in large groups of fully kindled rats to prove the reproducibility of its effect on ADT in individual animals. In a first study with 52 female kindled rats [20], a maximum tolerated dose of phenytoin (75 mg/kg i.p.) reproducibly increased the ADT in 21% (“phenytoin responders”), induced variable effects (i.e., an increase in one trial but no increase in another trial, or vice versa) in 58% (variable responders), and never increased the ADT in another 21% of kindled rats (“phenytoin nonresponders”). The difference in response to phenytoin between responders and nonresponders was dramatic in that average ADT increases induced by phenytoin in responders at plasma concentrations of 25–30 µg/mL were between 400% and more than 1000% above individual predrug control values, whereas no increase or even slight decreases in ADT were determined in nonresponders at the same phenytoin plasma concentration range. Interestingly, phenytoin responders and nonresponders did not differ in kindling acquisition or the severity and duration of their fully kindled seizures. In other words, there was no prognostic measure by which it could be predicted if a given kindled rat would respond or not respond to treatment with phenytoin [20]. The only significant difference between phenytoin responders and nonresponders was a higher control ADT in the kindled nonresponders (104 ± 22 µA) compared with the responders (56.5 ± 12 µM; *p* < 0.05). Interestingly, the difference in the response of kindled rats to phenytoin was restricted to kindled seizures because phenytoin induced the same anti-seizure effect on the threshold for generalized tonic electroconvulsions (determined via transauricular electrodes) in both groups of kindled rats [20]. Löscher and Rundfeldt [20] suggested that kindled rats with phenytoin resistance are a unique resource for the investigation of mechanisms of drug resistance in epilepsy, particularly because pathophysiological processes in phenytoin-resistant rats can be directly compared with those of kindled rats which reproducibly respond to this drug.

When Löscher and Rundfeldt [20] first reported the selection of phenytoin responders and nonresponders from large groups of amygdala-kindled Wistar rats, several groups that used this model for drug studies did not believe the findings but argued that they did not observe such striking inter-rat differences in drug response. However, Löscher’s group was the first to systematically search for inter-individual differences in ASM responses in large groups of amygdala-kindled rats, whereas all other research groups averaged drug responses in relatively small groups (*n* = 6–10). Indeed, previous studies on the effects of phenytoin in small groups of amygdala-kindled rats had yielded equivocal results, in that some groups reported anti-seizure effects at non-toxic doses [40,45,46,47] or anti-seizure effects only at large, ataxiogenic doses [48,49,50,51], while others found the drug to be only weakly active, inactive, or even proconvulsant [52,53,54]. One major difference between several studies in which phenytoin exerted anti-seizure effects and those in which the drug was ineffective was the intensity of electrical current used for evoking kindled seizures in fully kindled rats, which was the reason why Löscher’s group determined the effect of this drug on the individual seizure threshold (ADT) in their first [20] and all subsequent studies. Another important difference was the rat strain used (see Section 2.3.2).

In several subsequently performed prospective studies, Löscher’s group reproduced the original findings, although the frequency of responders and nonresponders slightly differed between studies. Average data from more than 200 rats showed a consistent anti-seizure response to a maximum tolerated dose of phenytoin (or its prodrug fosphenytoin) in only 16% of the animals, no anti-seizure response in 23%, and a variable response in the remaining 61% [18,30,55]. These differences were not due to pharmacokinetic issues, as plasma drug analyses demonstrated that all rats had phenytoin levels within the therapeutic concentration range (see Figure 3B). Furthermore, the subgroup selection did not depend on sex or estrous cycle, as phenytoin responders and nonresponders could also be selected from male Wistar rats [56].

A schematic illustration of the experimental protocol used for the selection of responders and nonresponders is shown in Figure 3A together with an example for ADT data and plasma drug levels from such subgroups (Figure 3B). Based on these data, Löscher suggested that the three subgroups of amygdala-kindled rats model three different clinical scenarios. The nonresponder subgroup models drug-resistant patients with TLE in which ASM treatment does not significantly reduce seizure frequency. The variable responder group models patients in which ASM treatment reduces seizure frequency but does not achieve complete control of seizures even at poorly tolerated doses. The responder subgroup models patients that achieve complete control of seizures during ASM treatment. Overall, Löscher concluded that kindled rats with phenytoin are a unique resource for the investigation of mechanisms for drug resistance in epilepsy, particularly because pathophysiological processes in phenytoin-resistant rats can be directly compared with those of kindled rats that reproducibly respond to this drug [30].

#### 2.3.1. The Resistance to Phenytoin Extends to Other ASMs

In various subsequent studies, Löscher’s group demonstrated that the resistance to phenytoin in nonresponders extends to several other ASMs and explored the mechanisms underlying this resistance [3]. In all the subsequent studies described in the following, only the responder and nonresponder subgroups of kindled Wistar rats were used.

In short, nine ASMs were tested in phenytoin-responding and nonresponding kindled rats. As shown in Figure 4A, most ASMs were significantly less efficacious or not efficacious at all in phenytoin nonresponders compared with phenytoin responders, demonstrating that phenytoin resistance in a subgroup of kindled Wistar rats extends to various other old and new ASMs [31]. This reflects the clinical situation in patients with TLE because most patients who are resistant to one ASM are also resistant to other ASMs, including newly developed drugs [13]. The only exception was levetiracetam, which exhibited the same efficacy in both subgroups of kindled rats (Figure 4A). The high efficacy of levetiracetam in this model is interesting and may either indicate that the relevant mechanisms of action of this ASM are not affected in phenytoin nonresponders (but see Section 2.3.3) or that the increased expression of multidrug transporters seen in this model (see below) does not affect brain levels of levetiracetam, which has been substantiated by microdialysis experiments in rats [57].

#### 2.3.2. Cellular and Molecular Mechanisms of Pharmacoresistance in Amygdala-Kindled Rats—Studies from the Löscher Group

Various factors that could be important for the presence of pharmacoresistant subgroups of amygdala-kindled Wistar rats were explored by Löscher’s group [3,58]. These studies showed that pharmacoresistance is not due to differences in the location of the kindling electrode in the amygdala, drug pharmacokinetics, seasonal variations in drug response, or the sex of the animals (Figure 4B), i.e., phenytoin nonresponders could also be selected from male Wistar rats.

Kindled phenytoin responders did not show more marked ataxia (or sedation) after injection of phenytoin than nonresponders, which did not point to a general, genetically determined subsensitivity of nonresponders to the pharmacodynamic effects of phenytoin [20]. To examine if the differences in sensitivity of kindled rats to the anti-seizure effects of phenytoin had a genetic background, the effect of phenytoin (15 mg/kg i.p.) on the threshold for electrically induced tonic extension seizures (MES threshold) was determined [20]. Phenytoin induced the same ~3-fold threshold increase in responders and nonresponders, thus suggesting that the resistance of kindled seizures to phenytoin in nonresponders did not relate to a low susceptibility of the animals to anti-seizure effects of phenytoin in general, but was restricted to kindled seizures, possibly related to differences between nonresponders and responders in neurophysiological or neurochemical alterations in the kindled focus.

To further examine the possible involvement of genetics in the differential effects of phenytoin found in kindled rats, Löscher’s group undertook breeding studies with phenytoin responders and nonresponders, using male and female Wistar rats [59]. Altogether, four generations of kindled Wistar rats were studied. The data suggested that the ability of kindled rats to respond or not to respond with an ADT increase after injection of phenytoin is genetically determined, although it does not follow a simple scheme of inheritance [59].

The involvement of genetics in the different drug sensitivity in responders and nonresponders selected from kindled Wistar rats was substantiated by a study in which Löscher and colleagues tested phenytoin in six other outbred and inbred rat strains, i.e., Sprague–Dawley, Wistar–Kyoto, Lewis, Fischer 344, ACI, and Brown Norway [60,61]. The only strain in which nonresponders could be selected was Brown Norway. However, in contrast to Wistar outbred rats, no responders could be selected in the Brown Norway strain, so Wistar rats were the only strain allowing the selection of nonresponders and responders. All subsequent experiments were thus conducted in the Wistar strain.

Besides genetics, another possible explanation for the development of different pharmacosensitivity in kindled rats would be the kindling process itself. Epileptic patients often initially respond to an ASM, but this effect may be lost with increasing duration of the disease, i.e., when epilepsy becomes chronic [10]. To address the influence of kindling on the anti-seizure response to phenytoin, Löscher’s group tested phenytoin’s anti-seizure effect on ADT before and after kindling in the same Wistar rats. Following kindling, rats were repeatedly tested with phenytoin to allow subgroup selection. Unexpectedly, in rats that were nonresponders after kindling, phenytoin exerted a significant anti-seizure effect before kindling [62]. This study thus indicated that kindled phenytoin nonresponders become nonresponders, at least in part, through the kindling process, i.e., the kindling-induced brain alterations. Because of the results of breeding studies in Wistar rats [59], the genetic background of an individual rat seems to determine whether it becomes a responder or nonresponder by kindling-induced limbic epileptogenesis.

This prompted a large study in cooperation with UCB Pharma in which the expression of ~5000 genes was compared between phenytoin responders and nonresponders. About 50 genes exhibited inter-group differences in expression, including the sodium channel beta 3 subunit and the delta subunit of the GABA_A_ receptor (unpublished findings).

Concerning the potential role of target alterations in pharmacoresistance [63], amygdala-kindled rats were selected for their in vivo anti-seizure response to phenytoin and divided into responders and nonresponders and then evaluated by comparing phenytoin’s effect on voltage-activated sodium currents in CA1 neurons [64]. Furthermore, given the potential role of calcium current modulation in the anti-seizure action of phenytoin, the effect of phenytoin on high-voltage-activated calcium currents was studied in CA1 neurons. Electrode-implanted but not kindled rats were used as sham controls for comparison with the kindled rats. In all experiments, the interval between the last kindled seizure and ion channel measurements was at least 5 weeks. In kindled rats with in vivo resistance to the anti-seizure effect of phenytoin (phenytoin nonresponders), in vitro modulation of sodium and calcium currents by phenytoin in hippocampal CA1 neurons did not differ from respective data obtained in phenytoin responders, i.e., phenytoin resistance was not associated with a changed modulation of the sodium or calcium currents by this drug. Compared with sham controls, phenytoin’s inhibitory effect on sodium currents was significantly reduced by kindling without difference between the responder and nonresponder subgroups. These findings seem to exclude the possibility that the phenytoin resistance is solely related to a reduced pharmacological sensitivity of sodium or calcium currents in nonresponders.

Another potential mechanism underlying medical intractability is that ASMs do not reach sufficiently high brain levels despite adequate plasma levels within the “therapeutic range”. This could be due to the overexpression of drug efflux transporters such as P-glycoprotein (Pgp; ABCB1) at the BBB [65]. Phenytoin is a Pgp substrate [2,66], so whether the expression of Pgp is enhanced by kindling and whether differences in Pgp expression exist between phenytoin responders and nonresponders in kindled Wistar rat populations were studied. Löscher’s group found that kindled rats have lower extracellular brain levels of phenytoin than age-matched controls [67] and that phenytoin-nonresponders differ from responders in a marked overexpression of Pgp in the kindled amygdala [68]. Thus, at least in part, overexpression of Pgp could be involved in the multidrug resistance in phenytoin-resistant kindled Wistar rats. Figure 5 summarizes the most important differences between phenytoin responders and nonresponders that Löscher’s group has found in the kindling model and how they translate to respective changes in patients with pharmacoresistant TLE.

In 2011, Löscher’s group tested whether individual amygdala-kindled Wistar rats also differ in their anti-seizure response to valproate and which mechanism may underlie the different responses to this major ASM [69]. It was found that good and poor valproate responders could be selected in groups of kindled rats by repeatedly determining the effect of valproate (200 mg/kg i.p.) on the ADT. Furthermore, there was a significant correlation between the anti-seizure response to valproate in kindled rats and its effect on the firing rate and pattern of GABAergic neurons in substantia nigra pars reticulata (SNR), a main basal ganglia output structure involved in seizure propagation, seizure control, and epilepsy-induced neuroplasticity [70]. The less valproate was able to raise the seizure threshold, the lower the valproate-induced reduction in the SNR firing rate and the valproate-induced regularity of SNR firing [69]. These data demonstrated for the first time an involvement of the SNR in pharmacoresistant experimental epilepsy and emphasized the relevance of the basal ganglia as target structures for new treatment options.

#### 2.3.3. Cellular and Molecular Mechanisms of Pharmacoresistance in Amygdala-Kindled Rats—Studies from Other Groups

Since Löscher’s group first characterized phenytoin responders and nonresponders in the amygdala kindling model in Wistar rats, several other groups have used this model as well, reproducing the findings of Löscher’s group with phenytoin, and reporting interesting cellular and molecular differences between the two subgroups. Indeed, Löscher’s model has become the most intensively studied preclinical tool for discovering potential mechanisms of ASM resistance.

As described in Section 2.3.2, one of the findings of Löscher’s group in this model was that the expression of the efflux transporter Pgp is significantly higher in the focus (the amygdala) of phenytoin nonresponders when compared with responders [68]. Subsequent experiments by another group in male Sprague–Dawley rats showed that the increased Pgp expression in phenytoin nonresponders extended to the hippocampus was associated with significantly reduced brain levels of phenytoin and carbamazepine (measured by microdialysis) and that this reduction in ASM levels could be counteracted by inhibiting Pgp with verapamil [71].

Chen et al. [72] reported that the drug efflux transporter multidrug resistance protein 1 (MRP1; ABCC1) is significantly upregulated in the cortex and hippocampus of phenytoin nonresponders. Brain microdialysis studies showed that the concentrations of phenytoin and carbamazepine in the cortical extracellular fluid were significantly decreased in the kindled phenytoin nonresponders. Pre-administration of probenecid restored drug levels back to their control levels [72]. Thus, both Pgp and MRP1 seem to be involved in the drug resistance of the phenytoin nonresponders.

Other groups reported differences in the expression of mitochondrial proteins in the hippocampus between phenytoin responders and nonresponders [73], enhanced synaptic vesicle trafficking in the hippocampus of phenytoin-resistant rats [74], and alterations of glutamate and GABA release in the hippocampus of phenytoin-nonresponders that resembled those found in patients with drug-resistant TLE [75]. Wang et al. [76] and Shi et al. [77] reported a significant decrease in hippocampal mRNA and protein levels of synaptic vesicle glycoprotein 2A (SV2A) in phenytoin nonresponders vs. responders, which corresponds to the reduction in SV2A in patients with drug-resistant mesial TLE (mTLE) [78,79]. SV2A is an integral constituent of synaptic vesicle membranes and has been demonstrated to be involved in vesicle trafficking and exocytosis processes crucial for neurotransmission [37]. Furthermore, SV2A is thought to be the main target of the ASMs levetiracetam and brivaracetam [37]. In amygdala-kindled rats, hippocampal low-frequency stimulation for two weeks was shown to increase SV2A levels and exert anti-seizure effects in phenytoin nonresponders [76], which would be consistent with the anti-seizure activity of hippocampal deep brain stimulation (DBS) in patients with drug-resistant TLE [80]. Wu et al. [81,82] reported that an increase in the expression of GABA_A_ and GABA_B_ receptors may be involved in the anti-seizure effects of hippocampal low-frequency stimulation in phenytoin nonresponders. Xu et al. (2018) reported that the hippocampal expression levels of the GABA_A_ receptor α1 subunit decreased significantly in phenytoin nonresponders, while levels of ICER (inducible cAMP early repressor) and BDNF (brain-derived neurotrophic factor) were increased. The changes in the GABA_A_ receptor α1, ICER, and BDNF were counteracted by hippocampal low-frequency stimulation [83].

Amygdala kindling is known to induce only discrete morphological changes in the brain [33]. The most prominent cellular effect is aberrant mossy fiber sprouting (MFS) in the dentate gyrus, i.e., the induction of sprouting of collaterals from the mossy fibers which terminate back upon the cells of origin in the dentate gyrus [84]. Sutula et al. [84] speculated that this would increase the reactivity of the dentate gyrus leading to increased amplitude responses which could be further amplified at each subsequent synaptic stage within the hippocampus. However, hippocampal lesions have little effect on kindling from the amygdala [33]. Consequently, MFS cannot be critical for the kindling effect in general, although it might reflect the mechanisms involved in hippocampal participation in kindling. Using transmission electron microscopy, Huang et al. [85] recently reported ultrastructural alterations of hippocampal neurons (cell body pyknosis, cell edema, rupture of the protruding membrane, blurred membrane structure before and after synapses, decreased synaptic vesicles, and thinned postsynaptic densities) in phenytoin nonresponders that were not observed in phenytoin responders. To the best of our knowledge, this is the first evidence that suggests that structural alterations are involved in drug resistance in the kindling model, which would be in line with the role of structural hippocampal alterations for ASM resistance in patients with TLE. Indeed, hippocampal cell loss, gliosis, and aberrant MFS are pathological hallmarks of drug-resistant TLE [2]. Using immunofluorescence and Western blot analysis, Huang et al. (2020) showed that the expressions of zinc transporter 3 (ZnT3) and glial fibrillary acidic protein (GFAP), i.e., markers of MFS and astrocyte proliferation, were markedly higher in phenytoin-resistant rats than responsive rats.

Using female Wistar rats as in Löscher’s initial studies, Xu et al. [86] reported that the expression of activated caspase-1 in the subiculum, but not the CA1, was upregulated in phenytoin-resistant amygdala-kindled rats. Genetic ablation of caspase-1 interfered with the genesis of pharmacoresistance in the kindling model. The pro-pharmacoresistance effect of subicular caspase-1 was mediated by its downstream inflammasome-dependent interleukin-1β. Administration of CZL80, a small molecular caspase-1 inhibitor, attenuated seizures in phenytoin nonresponders and decreased the neuronal excitability in brain slices obtained from patients with pharmacoresistant TLE. Data in the kindling model were confirmed in the IHK model of TLE in Wistar rats. The data of Xu et al. [86] suggest that the subicular caspase-1-interleukin-1β inflammatory pathway may play a critical role in drug-resistant TLE and that caspase-1 may be a potential target for more effective therapies.

#### 2.3.4. Advantages and Limitations of Phenytoin-Resistant Kindled Rats as a Model of Drug-Resistant Epilepsy

A major advantage of the phenytoin-resistant kindled rat model is that it allows a direct comparison between ASM-resistant and ASM-responsive rats from the same rat strain, which is an important benefit for studies on mechanisms of pharmacoresistance. However, because rats have to be implanted with electrodes, followed by subsequent kindling, and then several weeks of repeated testing of phenytoin’s effects on ADT before ASM-resistant and ASM-responsive rats are obtained for further studies (Figure 3A), the model is very time and labor intensive. On the other hand, because seizures are induced at will, the model does not require any continuous EEG/video monitoring of seizures. One may argue that the lack of SRSs is a major limitation of the amygdala kindling model and that kindling, at best, is only a model of partial epileptogenesis. Indeed, after about 50–100 once-daily amygdala stimulations, kindled rats start to develop SRSs [87]. However, the pharmacology of induced kindled seizures in fully kindled rats and SRSs in post-status models of TLE is remarkably similar [88], thus indicating that models with SRSs have no general advantage in this regard. Nevertheless, as discussed below, models with SRSs may allow discovery of mechanisms of drug resistance that are not accessible by kindling, such as the role of overt neuronal damage, which does not occur in the kindling model.

In conclusion, since Löscher’s group first characterized this model in Wistar rats, several other groups have used this model as well and reported highly interesting findings on potential cellular and molecular mechanisms of ASM resistance and potential therapeutic targets (see Section 2.3.3).

### 2.4. Post-Status Epilepticus Models of Mesial Temporal Lobe Epilepsy

In the kindling model, seizures are elicited by electrical stimulation for drug testing, i.e., the animals do not exhibit chronic epilepsy with SRSs. Although it is not known whether this is a disadvantage for evaluating the therapeutic potential of novel compounds for the treatment of epilepsy or for studying mechanisms of drug resistance, a model of DRE mimicking the human condition, i.e., inadequate control of spontaneous seizures, would certainly be an important development. This prompted Löscher’s group to study the individual response to ASMs in TLE models in which SE is either induced electrically (by sustained electrical stimulation of the BLA) or chemically, which, depending on SE duration and severity, leads to the development of SRSs. Such post-SE models of TLE are widely used in epilepsy research, particularly for studying epileptogenesis after brain injury and for testing potential antiepileptogenic treatments [17,89,90,91]. In addition, more recently, post-SE models of TLE have been increasingly used to study the pharmacology of the SRSs occurring in these models. However, to the best of our knowledge, Löscher’s group was the first to examine whether ASM responders and nonresponders can be selected from such models, similar to the previous findings in the kindling model. This led to the discovery of striking differences in the response of rats with SRSs to the ASM phenobarbital [21,23].

### 2.5. Selection of Phenobarbital Responders and Nonresponders from Epileptic Rats in the BLA-SE Model of TLE

Based on the previous experience of Löscher’s group in different rat strains and sexes, female outbred rats of the Sprague–Dawley strain were used for drug studies in this model because prolonged (~25 min) electrical stimulation of the BLA results in most of these rats in a self-sustained generalized convulsive SE which induces the development of SRSs in >90% of the animals [92]. We have not yet evaluated whether male Sprague–Dawley rats differ from females in this model. The female rats were stereotactically implanted with a bipolar electrode into the right BLA in the same way as for amygdala kindling [92]. About two weeks after electrode implantation, the rats were electrically stimulated via the BLA electrode for induction of a self-sustained SE. With the chosen stimulation parameters, about 90% of rats developed a self-sustained SE with generalized convulsive seizures. After 4 h, SE was interrupted by diazepam (10 mg/kg i.p.); if necessary, the application of this dose of diazepam was repeated, but in most rats seizure activity was terminated after the first diazepam injection. Starting about four weeks later, the rats were monitored by EEG video recordings for up to two months until the first SRSs were detected [21,92]. Rats with frequent SRSs were used for drug studies.

To evaluate whether epileptic rats from this model can be segregated into drug-refractory and drug-responsive animals by ASM testing, a first study was performed with phenobarbital in a group of 11 rats with SRSs [21]. Phenobarbital was chosen because it is an efficacious ASM in rat models of TLE with a sufficiently long half-life in rats (~12 h) to allow maintenance of “therapeutic” drug levels during prolonged treatment [93]. As described in detail previously [21], several preliminary experiments were performed with phenobarbital to develop a dosing protocol allowing to maintain plasma drug concentrations within or above the therapeutic range (10–40 µg/mL) over 24 h/day, 7 days/week. Furthermore, Löscher’s group was interested in administering phenobarbital at maximum tolerated doses, so rats were closely observed for adverse effects. Based on these preliminary experiments, a dosing protocol with an i.p. bolus dose of 25 mg/kg in the morning of the first treatment day, followed 10 h later by an administration of 15 mg/kg, and then twice daily 15 mg/kg for the 13 subsequent days, was used in rats with SRSs. Before the onset of drug treatment, baseline seizure frequency was determined over two weeks (predrug control period); then, phenobarbital was administered over two weeks, followed by a postdrug control period of two weeks. Blood was sampled 10 h after the first drug injection and 12 h after the last drug injection for drug analysis in plasma. In each of the 6 weeks of the experiment, seizures were continuously (24 h/day, 7 days/week) monitored by video EEG recording as described in detail recently [21]. A schematic illustration of the drug trial is shown in Figure 6A.

Similar to Löscher’s studies with phenytoin in amygdala-kindled Wistar rats, phenobarbital responders and nonresponders could be selected in epileptic Wistar rats [21]. This finding was reproduced by several subsequent prospective studies and the results of these studies were combined for the data in Figure 6B. Overall, 61% of the rats were responders and 39% were nonresponders. Similar figures were reported when the experiments were performed by the group of H. Potschka in Munich, Germany [17], demonstrating that the finding was reproducible in different laboratories. All rats received phenobarbital at maximum tolerated doses as indicated by the marked sedation that was observed in all rats during treatment. Analysis of plasma drug concentrations showed that drug concentrations within the therapeutic range (10–40 µg/mL) were maintained in all rats throughout treatment (Figure 6B). Most responders exhibited a complete suppression of seizures, but a reduction in seizure frequency of 50–75% was also accepted as a response. Nonresponders often exhibited an increase in seizure frequency (Figure 6B).

The type of SRSs was the same in all responders and nonresponders, i.e., generalized convulsive seizures, resembling stage 4 or 5 seizures on the Racine scale [94]. Furthermore, the severity of the initial, electrically-induced SE was not different between responders and nonresponders, indicating that the same severity and duration of SE produces two subgroups of epileptic rats, ASM responders, and nonresponders.

#### 2.5.1. Extension of Resistance to Other ASMs

In subsequent studies, Löscher’s group reported that the lack of response to phenobarbital extends to phenytoin [22], so this model fulfills the minimum requirements of an animal model of drug-refractory epilepsy, that is, persistent seizure activity not responding to at least two ASMs at maximum-tolerated doses [8]. Lamotrigine proved to be more effective than phenytoin in phenobarbital nonresponders [24].

#### 2.5.2. Mechanisms of Resistance

Figure 5 summarizes the various studies that Löscher’s group undertook to identify mechanisms that could explain the resistance to phenobarbital in nonresponders. Interestingly, the average seizure frequency of phenobarbital nonresponders was significantly higher than that of responders [95], which is in line with clinical experience that the frequency of seizures in the early phase of epilepsy is a dominant risk factor that predicts refractoriness [96]. However, resistance to treatment also occurred in rats that did not differ in seizure frequency from responders, indicating that disease severity alone is not sufficient to explain ASM resistance [95].

In addition to the difference in average seizure frequency, it was found that phenobarbital-nonresponders differ from responders in behavioral and cognitive alterations [97]. Furthermore, Löscher’s group found that the majority (90%) of phenobarbital-nonresponders exhibit hippocampal damage, whereas such damage was determined in only 7% of responders, so neuron loss in the hippocampus, particularly in the dentate hilus, is a characteristic feature of phenobarbital-resistant rats [98,99]. Again, this observation in the rat model is in line with clinical experience, in that hippocampal sclerosis in patients with TLE carries a poor prognosis [100].

In addition to hippocampal damage, phenobarbital nonresponders differed from responders in increased expression of the efflux transporter Pgp in the hippocampus and parahippocampal areas [101] and altered subunit expression and binding characteristics of GABA_A_ receptors in the hippocampus [98,99], which may be critically involved in the lack of anti-seizure efficacy of phenobarbital in nonresponders. These clear differences between phenobarbital responders and nonresponders also indicate that the strict definition of response that Löscher’s group chose for selection was suitable for differentiating between pharmacoresistant and pharmacoresponsive rats.

Interestingly, the only comparable difference between responders and nonresponders in the two models (kindling, post-SE TLE) explored by Löscher’s group was increased expression of Pgp at the BBB of nonresponders (Figure 5). To directly address the possibility that Pgp is critically involved in ASM resistance, Löscher’s group performed a proof-of-concept experiment with the Pgp inhibitor tariquidar in phenobarbital-resistant epileptic rats [102]. Coadministration of tariquidar fully restored the anti-seizure activity of phenobarbital without altering plasma pharmacokinetics or neurotoxicity of the ASM, demonstrating that inhibiting Pgp in epileptic rats with proven drug resistance counteracts resistance [102].

### 2.6. Selection of ASM Responders and Nonresponders from Epileptic Rats and Mice in the Pilocarpine Model of TLE

Prompted by these studies in the BLA model of post-SE TLE, Löscher’s group performed similar experiments in the pilocarpine model of TLE, in which SE is induced by systemic administration of the cholinergic agonist pilocarpine. Epileptic rats were either treated with levetiracetam [103] or phenobarbital [23]. In the study with levetiracetam, which, because of its short half-life in rats [93], was administered continuously via subcutaneously implanted osmotic minipumps in female Wistar rats, 38% of the rats were responders with complete or almost complete seizure control, another 38% were nonresponders, while the remaining rats could not clearly be included in either group because of variation between pre- and postdrug control seizure frequency [103]. In the study with phenobarbital in the pilocarpine model in female Sprague–Dawley rats, in which this ASM was given with the same dosing protocol as used for the BLA model of post-SE TLE (see above), 50% of the epileptic rats did not adequately respond to this drug, whereas phenobarbital significantly decreased seizure frequency and severity in another 50% of the animals [23]. Responders and nonresponders did not differ in predrug seizure frequency, drug plasma levels, or hippocampal neurodegeneration, but behavioral differences were observed in anxiety tests. These findings indicate that in the pilocarpine model, similar to epilepsy patients, epileptic rats of different strains differ in their response to ASMs, which is most likely due to as yet unknown genetic or epigenetic factors.

Prompted by Löscher’s studies on the selection of ASM responders and nonresponders from large groups of epileptic rats, Moon et al. [104] examined whether such subgroups can also be selected in the mouse pilocarpine model of TLE. For this purpose, a group of 37 male epileptic C57BL/6J mice with SRSs were treated with levetiracetam (15 mg/kg i.p. twice daily for 7 days) and, after a 7-day washout, those mice that were resistant to levetiracetam were treated with valproate (300 mg/kg, twice a day for 7 days). SRSs were recorded by video EEG monitoring. The drug response was defined by >75% suppression of SRSs compared with vehicle baseline control. Among the 37 epileptic mice with SRSs, 23 showed >75% fewer SRSs during the administration of levetiracetam, while 14 (38%) were resistant to this ASM. In 7 of the 14 levetiracetam nonresponders, the resistance extended to valproate (multidrug-resistant group), while in the other 7 mice, valproate was effective (levetiracetam resistant/valproate sensitive group). Multidrug-resistant mice displayed distinctive behaviors in the object exploration and elevated plus- maze tests, which were not observed in the levetiracetam responders. The expression of four microRNAs (miR-206, miR-374, miR-468, and miR-142-5p) was differently modulated in the multidrug-resistant versus both control and levetiracetam responder groups [104]. The authors hypothesized that modulation of the identified miRNAs may play a key role in developing pharmacoresistance and behavioral alterations in multidrug-resistant epileptic mice. Indeed, miRNAs, a class of short noncoding RNAs negatively regulating gene expression, have emerged in recent years in the development of ASM resistance [105]. In the systemic (i.p.) kainate model of TLE in mice (see Section 2.7), ASM resistance could be suppressed using miR-221-3p mimics [106].

### 2.7. Selection of ASM Responders and Nonresponders from Epileptic Mice in the Kainate Model of TLE

In the IHK model of mTLE (see Section 4.1.1), Löscher’s group examined the anti-seizure effects of 6 ASMs on focal electrographic SRSs and secondarily generalized convulsive SRSs in female epileptic FVB/N mice, showing that the highly frequent focal electrographic seizures were resistant to carbamazepine and phenytoin, whereas valproate and levetiracetam exerted moderate and phenobarbital and diazepam marked anti-seizure effects [25]. All ASMs seemed to suppress the infrequent generalized convulsive seizures. Next, Löscher’s group investigated the inter-individual variation in the anti-seizure effects of these ASMs and, in the case of electrographic seizures, found responders and nonresponders to all ASMs except carbamazepine. Most nonresponders were resistant to more than one ASM [25]. In a subsequent study, Bankstahl et al. [107] reported that knockout of *ABCB1* does not alter the efficacy of ASMs against electrographic seizures in the IHK mouse model, indicating that Pgp is not involved in the mechanisms explaining that focal electrographic seizures are resistant to some ASMs in this model. This was substantiated by the finding that epileptic wild-type mice do not exhibit increased Pgp expression in this model.

Using systemic (i.p.) administration of kainate to induce SE and subsequent SRSs in male C57BL/6 mice, Fu et al. [106] reported that ASM responders and nonresponders can be selected by valproate. A total of 48 epileptic mice were treated with valproate (150 mg/kg, p.o.) twice daily for 4 weeks. A total of 25 of the mice exhibited a striking seizure-suppressive effect of valproate (responders), while another 25% did not differ in SRS frequency from kainate controls (nonresponders). Through proteomics analysis, Fu et al. (2021) identified that hypoxia-inducible factor (HIF)-1α, an essential effector molecule of hypoxia and inflammation, was overexpressed in the valproate nonresponders, and regulated the expression of interleukin-1β and tumor necrosis factor-α. Increased expression of HIF-1α led to the increase in microglia and induced their polarization from the M2 phenotype to the M1 phenotype, which triggered the release of proinflammatory mediators. Bioinformatics analysis of public databases demonstrated that miR-221-3p was reduced in valproate nonresponders and negatively regulated HIF-1α expression. Intervention using miR-221-3p mimics reduced HIF-1α expression markedly and suppressed the activation of microglia and the release of inflammatory mediators, which reduced epileptic seizures in the valproate nonresponders [106]. These effects were also seen with a HIF-1α inhibitor, further substantiating the role of neuroinflammation in ASM resistance.

### 2.8. Advantages and Limitations of ASM-Resistant Epileptic Rats and Mice as a Model of Drug-Resistant Epilepsy

As shown by the drug studies of Löscher’s group in chemical and electrical post-SE models of TLE, it is possible to select animals with ASM-refractory or ASM-responsive spontaneous seizures from such models. However, several important issues have to be dealt with in such studies. To avoid rats being falsely considered drug-refractory because of pharmacokinetic factors (e.g., too rapid drug elimination), drug administration protocols have to be based on ASM pharmacokinetics in rats, so ASM levels are maintained in plasma (and brain) throughout the treatment period [93]. Female rats are known to eliminate several ASMs more slowly than males, which is an advantage for chronic drug studies [42]. Löscher’s group has also shown that the estrous cycle is not a bias in such studies and that neither seizure threshold nor ASM response is affected by the estrous cycle, at least under the experimental conditions of Löscher’s studies [44,108,109,110]. Interpretation of results from drug testing in rats with SRSs can be performed in a similar way as conducted in patients with epilepsy with endpoints such as seizure frequency and seizure severity. However, because of the need to continuously monitor SRSs over several weeks (Figure 6), drug testing in post-SE models of TLE is technically difficult, time consuming, and expensive. Automated seizure recording systems, including EEG monitoring via miniaturized telemetric devices, and new methods of prolonged drug delivery, will hopefully soon make chronic studies in such models less labor-dependent and increase the use of epileptic animals for evaluation of drug effects. However, our own experience with different algorithms for automated seizure detection in the EEG of epileptic rodents is as yet not quite promising, so Löscher’s group still prefers visual EEG and video analyses.

## 3. Animal Models with Induced Seizures That Are Resistant to ASMs

Apart from the approach of Löscher’s group to select ASM-resistant and responsive rats from different models of TLE and then use these two subgroups for determining potential cellular and molecular mechanisms of drug resistance, a variety of TLE models are used in the search for more effective treatments of drug-resistant seizures. For this aim, rats and mice are not individualized in their response to treatment but group data are used for response analyses. An important example of such an approach is the Epilepsy Therapy Screening Program (ETSP; previously the Anticonvulsant Screening Program) of the National Institute of Neurological Disorders and Stroke (NINDS) in the U.S. [111]. The mission of this program is to facilitate the discovery of new therapeutic agents that address unmet medical needs in epilepsy, i.e., drug resistance and disease prevention or modification. The program provides opportunities for researchers from academia and industry in the U.S. and abroad to submit compounds for testing in a battery of well-established rodent seizure and epilepsy models, thus assembling compelling efficacy packages that serve to facilitate the advancement of new compounds toward the clinic for the symptomatic control of seizures [112]. The current testing scheme for pharmacoresistant epilepsy is illustrated in Figure 7. As shown in this figure, the workflow of compound testing starts with an identification phase, including assays such as the MES test and the 6 Hz models (see Section 3.1) that allow for higher throughput. Because the 6 Hz models (44 mA in mice and 80 V in rats) are pharmacoresistant to several major ASMs (Figure 1), investigational compounds found to be effective in the 6 Hz models without significant tolerability issues may be advanced into the differentiation phase of the testing scheme. While the MES test is sensitive to numerous ASMs and is not considered a pharmacoresistant test, efficacy in only this model can indicate the activity of a compound that, combined with information from the compound supplier such as targets, shared under confidentiality with NINDS ETSP staff, might then be considered sufficiently novel to move into the differentiation phase [111,112].

As shown in Figure 7, during the identification phase, compounds are also assessed for the potential to induce impaired motor activity using multiple assays including a rotarod assay (see Figure 2A). In the latter test, the dose that produces behavioral impairment in 50% of the animals treated (TD_50_) is determined, which allows one to compare the dose that affords seizure protection in 50% of the animals treated, i.e., ED_50_, with the TD_50_ by calculating the “protective (or therapeutic) index” (PI: TD_50_/ED_50_) and thus define whether a compound confers seizure protection at doses that are not accompanied with motor impairment (i.e., PI >> 1).

Finally, more etiologically relevant assays during the identification phase include testing in corneal-kindled mice and, for those novel investigational compounds for which brain penetration might prove difficult, in a brain slice preparation that is obtained from kainate-treated rats (see Section 4.1.3) and exhibits recurrent epileptiform discharges in the entorhinal cortex, which are resistant to several ASMs [111].

Not all projects profiled in the DRE workflow shown in Figure 7 initiate with the acute seizure models. The starting point can depend on information the participant shares under confidentiality with the NINDS ETSP staff. Thus, as described by Kehne et al. [112], the workflow is flexible, and decisions regarding starting points and compound advancement are made by the NINDS ETSP staff based on testing results combined with the understanding of novel properties of the investigational compound shared under confidentiality by the supplier.

In the differentiation phase of the ETSP’s drug-resistant epilepsy workflow, several chronic TLE models with seizures that are resistant to several ASMs are used, including the lamotrigine-resistant amygdala-kindled rat (see Section 3.3), the IHK mouse model (Section 4.1.1), the intra-amygdala kainate (IAK) mouse model (Section 4.1.2), and a post-SE model of TLE in which SE is induced in rats by systemic administration of kainate (Section 4.1.3). Similar batteries of animal models of drug-resistant seizures are being used by other groups, including Steve White’s group at the University of Washington, Seattle. These models will be described in more detail in the following pages.

### 3.1. The 6 Hz Psychomotor Seizure Model

The 6 Hz psychomotor seizure is an acutely evoked seizure in mice [113] or rats [114] that has shown promise as a rapid throughput screening model of pharmacoresistant seizures. The 6 Hz model is based on the fact that electrical stimulation by low-frequency (6 Hz) rectangular pulses of 0.2 ms duration delivered through corneal electrodes for a relatively long duration (3 s) induces less seizure spread than the widely used high-frequency (50 or 60 Hz) MES model [113]. Once evoked, the seizure is characterized by a momentary stun, vibrissae twitching, unilateral or bilateral forelimb clonus, and Straub tail. In 1953, it was originally described as a model of ‘psychomotor seizures’ [115], but was abandoned as a screening model shortly after its description because of its relative lack of sensitivity to phenytoin, one of the predominately used ASMs at that time. It was this observation that led White and colleagues to reassess the 6 Hz seizure model in 2001 as a potential model for therapy-resistant seizures, using male CF-1 mice [113]. Results from these studies confirmed pharmacoresistance to phenytoin and extended the observation to include other sodium channel-blocking drugs including lamotrigine and carbamazepine, and the broad-spectrum ASM, topiramate. As was described by Barton et al. [113], there is often a shift in the potency of an ASM when the current intensity is increased above the current required to evoke a characteristic seizure in 97% of the mice stimulated, i.e., the CC_97_ (22 mA). For example, at this threshold stimulus intensity, many of the ASMs show efficacy; however, as the intensity is increased to 1.5×CC_97_ (32 mA) and 2×CC_97_ (44 mA), the potency and often the PI declines (see Section 7) [113]. A good example of this is provided by levetiracetam. In their study, Barton et al. (2001) demonstrated that levetiracetam was effective and reasonably potent (ED_50_: 4.6 mg/kg) at the CC_97_ stimulus intensity, but the potency decreased markedly as the intensity of the stimulation current was increased to 1.5×CC_97_ (ED_50_: 19.4 mg/kg) and 2×CC_97_ (ED_50_: 1089 mg/kg). In contrast to levetiracetam, the novel ASM, cenobamate, retained its potency in the 6 Hz seizure test in mice as the current intensity was increased from the CC_97_ (ED_50_: 11 mg/kg) to 1.5×CC_97_ (17.9) and 2×CC_97_ (16.5 mg/kg) [116]. Interestingly, in several clinical trials with cenobamate in patients with drug-resistant seizures, this drug provided high rates of seizure freedom, indicating that it may provide new hope for treatment-resistant epilepsy [117].

Figure 8 illustrates a comparison of i.p. ED_50_s of various ASMs determined in the MES vs. 6 Hz (32 mA and 44 mA) tests in male CF-1 mice. Furthermore, TD_50_s determined in the rotarod test are shown. As noted above, most sodium channel modulators are much more effective in the MES vs. 6 Hz tests, while the opposite is true for GABAergic drugs such as clonazepam, phenobarbital, and tiagabine. Levetiracetam is strikingly more potent in the 6 Hz (32 mA) than in the MES test. Thus, whereas this ASM is ineffective in the MES and PTZ tests [35], the 6 Hz (22 or 32 mA) model would have identified levetiracetam as a potentially useful ASM. Similarly, levetiracetam was found to be highly effective in blocking seizures in amygdala-kindled rats [35], which was an important reason for the further development of this drug in the 1990s [37,118]. The lack of efficacy of levetiracetam in traditional screening tests demonstrates the importance of utilizing models that more closely resemble the human pharmacological condition, which was one of the reasons to include 6 Hz and kindling models in the ETSP. The data in Figure 8 illustrate that higher potency in the 6 Hz vs. MES models as seen with levetiracetam is an exception and only shared by a few ASMs such as clonazepam and tiagabine.

In 2017, Metcalf and colleagues established a rat equivalent of the mouse 6 Hz test to expand the testing battery of the ETSP and provide another means to differentiate promising ASMs [114]. Phenotypically, the 6 Hz seizure in a rat looks similar to that observed in the mouse; however, some differences in the pharmacology of ASMs were noted. For example, the K^+^ channel activator, retigabine (ezogabine), was more effective in the rat model than in the mouse; whereas most of the Na^+^ channel modulating drugs including carbamazepine, lacosamide, and phenytoin displayed a larger PI in the mouse than the rat at the higher stimulus intensity of 2×CC_97_ (80 V). The GABA uptake inhibitor, tiagabine, also possessed a more favorable PI in the mouse than the rat at the higher stimulus intensity. What is clear, is that the 6 Hz test, whether it is conducted in the mouse or rat displays a degree of pharmacoresistance to the current standards of care and allows an investigator an opportunity to differentiate their investigational drug based on potency and efficacy by varying the stimulation strength as described above [113,114]. Whether efficacy testing in the 6 Hz test will identify a new generation of ASMs for pharmacoresistant epilepsy has yet to be established because no single drug has been brought to the clinic solely based on efficacy in this model. In the study of Barton et al. (2001) and in the ETSP, the CF-1 outbred mouse strain was used. In a subsequent study by Leclercq and Kaminski (2015a), it was shown that the genetic background of mice strongly influences treatment resistance in the 6 Hz seizure model. Thus, in contrast to CF-1 mice, NMRI mice are not resistant to phenytoin in the 6 Hz test. Similarly, NMRI mice are much more responsive to levetiracetam in this test than CF-1 (or C57BL/6) mice [119]. It is not known whether similar strain differences are present in the 6 Hz model in rats; the study of Metcalf et al. (2017) was performed in male Sprague–Dawley rats. In general, the rodent strain and sex, variables of the animals’ environment, and many other factors can affect the expression and pharmacological sensitivity of seizures in a variety of models, which has to be considered when performing drug studies [120].

### 3.2. Induction of Acute Seizures in Epileptic Animals

Traditionally, acute seizure tests, such as the MES, PTZ, and 6 Hz seizure models, have been performed in normal healthy (naive) rodents that do not exhibit any of the brain alterations that are found in chronic epilepsy and are likely involved in the mechanisms leading to intractable seizures. Although various animal models of chronic epilepsy with SRSs are available, drug testing on SRSs necessitates laborious video EEG seizure monitoring. In a landmark paper, Blanco et al. [121] suggested that instead of monitoring SRSs, chemical or electrical induction of acute seizures in epileptic rodents (in which epilepsy was induced by pilocarpine) may be used as a surrogate for testing the efficacy of novel ASMs against refractory SRSs, potentially identifying ASMs with new pharmacologic profiles. This idea stimulated other groups to compare ASM efficacy in acute seizure models in epileptic vs. nonepileptic mice to further explore the suggestion of Blanco et al. [121]. In the initial study of Blanco et al. [121], several ASMs (valproate, phenobarbital, and phenytoin) were shown to lose their efficacy on acute seizures, when such seizures were induced by PTZ in epileptic rather than nonepileptic rats, whereas this was not observed when using the MES test. Subsequent studies confirmed the loss of anti-seizure efficacy of valproate against PTZ-induced seizures in epileptic mice, but several other ASMs (phenobarbital, diazepam, lamotrigine, and levetiracetam) tended to be more effective against PTZ in epileptic than nonepileptic mice [122]. This was also observed when using the 6 Hz model of focal seizures in epileptic mice, in which the potency of levetiracetam, in particular, was markedly increased compared with nonepileptic animals [123,124]. When using the MES threshold for determining anti-seizure effects in epileptic and non-epileptic mice, Löscher’s group found that phenobarbital exhibited the same efficacy in both groups, but bumetanide and bumetanide derivatives potentiated the effect of phenobarbital only in epileptic mice [125]. Overall, these observations suggest that performing acute seizure tests in epileptic rodents provides valuable information on the pharmacological profile of ASMs. Löscher [126] suggested that the translational value of drug effects against acute seizures in epileptic rodents may be significantly higher than respective effects in nonepileptic mice or rats, particularly for drugs with mechanisms inherent to disease-induced brain alterations. This hypothesis needs to be further explored. However, overall, with some exceptions, the high degree of drug resistance of SRSs in some of the post-SE models of TLE discussed below is not reflected by chemical or electrical induction of acute seizures in epileptic animals of such models.

### 3.3. The Lamotrigine-Resistant Kindled Rat Model

As described in Section 2.1, kindling is the process that results when an animal is exposed to repeated subconvulsive electrical stimulations delivered to a limbic brain structure such as the amygdala, hippocampus, or piriform cortex [32]. With repeated stimulation, the initially subconvulsive current produces a focal seizure that eventually generalizes to produce a generalized tonic–clonic seizure. In contrast to the acute 6 Hz seizure model, the kindling model is much more labor-intensive and thus more expensive to conduct. However, kindling does represent a chronic model of network hyperexcitability and in this respect is more closely aligned to actual epilepsy; albeit, SRSs are rare in the kindling model unless the animal is “over-kindled” [127].

In the rat amygdala kindling model, it has been known for some time that exposure to a low dose of lamotrigine during the kindling process results in a state of pharmacoresistance to lamotrigine in the fully kindled animal [128]. Similarly, the same phenomenon has been observed in animals that were chemically kindled using the chemoconvulsant PTZ [129]. To develop the lamotrigine-resistant amygdala-kindled rat as a screening tool, subsequent investigations confirmed cross-tolerance to carbamazepine, but not valproate [28].

In 2019, Metcalf and colleagues further refined the pharmacological profile of the lamotrigine-resistant amygdala kindled rat to include the Na^+^ channel blockers eslicarbazepine, lacosamide, phenytoin, and rufinamide [130]. In contrast to the Na^+^ channel modulating drugs, ASMs that target GABA receptors (i.e., clobazam, clonazepam, and phenobarbital) and GABA uptake (tiagabine) were found to produce dose-dependent efficacy in the lamotrigine-resistant kindled rat [130]. Like the Na^+^ channel blockers, the T-type Ca^++^ channel blocker ethosuximide, the SV2A ligand, levetiracetam, and the broad-spectrum ASM, topiramate, were all without effect in the lamotrigine-resistant amygdala-kindled rat. In contrast, the Ca^++^ channel ɑ_2_δ modulator, gabapentin, displayed modest efficacy, and the K_v_7.2/3 activator, retigabine, and the multi-target ASM valproate were highly effective [130]. The only ASMs to display efficacy at doses devoid of behavioral impairment, i.e., PI ≥ 1, were those that increase GABA-mediated inhibition (phenobarbital and tiagabine), retigabine (K_v_7.2/7.3 modulator), and the broad-spectrum ASM, valproate. As would be expected from previous data of Löscher’s group on ASMs in amygdala-kindled rats (see Section 2.2), individual lamotrigine-resistant kindled rats varied widely in their response to a given ASM [130] but the sample size was too small to allow any conclusion about subgroups. Based on this comprehensive pharmacological profile, it is clear that the lamotrigine-resistant amygdala-kindled rat displays a profile consistent with multidrug pharmacoresistance that supports its utility as a moderate throughput screening model to further differentiate investigational ASMs. The model also highlights the impact of the development of tolerance (loss of efficacy) and cross-tolerance during prolonged treatment with an ASM (i.e., lamotrigine) as an important mechanism in drug resistance [131].

### 3.4. Corneal Kindling with 50 or 60 Hz in Mice and Rats

To develop a less labor-intensive mouse equivalent to the lamotrigine-resistant amygdala-kindled rat, Koneval et al. [132] characterized the pharmacological profile of male CF-1 mice kindled by repeated corneal stimulation (60 Hz for 3 s delivered through silver-coated corneal electrodes) in the presence of a low dose of lamotrigine. In their evaluation, they observed that corneal-kindled mice (CKM), kindled in the presence of a low dose of lamotrigine, develop a similar resistance to lamotrigine, carbamazepine, and phenytoin, whereas, in contrast to the lamotrigine-resistant amygdala-kindled rat, the lamotrigine-resistant CKM was resistant to the K_v_7.2/3 activator, retigabine, the benzodiazepine, diazepam, and to a modest extent, the broad-spectrum ASM, valproate [132]. These findings demonstrate that the lamotrigine-resistant CKM, much like the lamotrigine-resistant amygdala-kindled rat displays a pharmacological profile consistent with pharmacoresistance.

From a drug screening perspective, the lamotrigine-resistant CKM offers several advantages over the rat model. For example, mice, because of their lower body weight require less drug substance per dose compared with rats, no surgical implantation of kindling electrodes and post-surgery recovery period, and the kindling procedure itself is less technically demanding. The outcome measures are similar, i.e., the effect of a drug on seizure severity as measured by the Racine scale [94]. The one disadvantage of the CKM model is that there is no way, other than the Racine seizure score, to evaluate the effect of a drug on focal seizure activity. In contrast, in the amygdala-kindled LTG-resistant rat, one can utilize the electrical ADT and afterdischarge duration, in addition to the Racine seizure score as outcome measures for focal seizure severity.

In contrast to the approach used by Koneval et al. [132] with mice corneally kindled in the presence of a low dose of lamotrigine, CKM kindled in the absence of lamotrigine are quite sensitive to ASMs, demonstrating strong positive correlations with several diverse screening models [133,134]. Thus, such CKM cannot replace the amygdala kindling model that is per se resistant to various ASMs, as outlined in Section 2.2 [135]. Furthermore, in contrast to amygdala kindling, phenytoin proved highly potent and efficacious to block corneally kindled seizures and only one nonresponder could be selected out of 75 fully corneally kindled NMRI mice repeatedly tested with phenytoin [135]. However, like amygdala-kindled rats, CKM were highly susceptible to the anti-seizure effect of levetiracetam, indicating that corneal kindling of mice represents a sensitive and valid screening model for the identification of new therapies for focal epilepsy in man [133].

Interestingly, in contrast to the lamotrigine-resistant CKM, mice that were corneal kindled using 60 Hz of constant voltage kindling were not only resistant to lamotrigine and carbamazepine but also to phenobarbital [136]. The latter model is sensitive to the ASMs diazepam, levetiracetam, and valproate (Figure 1). The novel constant voltage CKM model is unique because it does not require prior ASM exposure (unlike the lamotrigine-resistant kindling models) and pharmacoresistance is likely a function of the seizure severity (intrinsic severity hypothesis) and/or changes in molecular target (albeit additional studies are required to confirm this hypothesis). Another advantage of this model is that the seizure phenotype of the 60 Hz constant voltage model is more consistent with that in the 6 Hz kindling paradigm in that the seizures present along the Racine scale and are therefore easier to score than in the often more complex 6 Hz kindling model [136].

In addition to mice, rats have been corneally kindled and used for ASM testing [137,138]. However, this model has no apparent advantages to the corneally kindled mouse.

### 3.5. Corneal Kindling with 6 Hz in Mice

In the traditional corneal kindling model in mice, a low-intensity current (2–3 mA) 50 Hz (or 60 Hz) electrical stimulation is applied twice daily for 3 s via corneal electrodes, which results in a fully kindled state within about 2 weeks [133,134,135]. In 2014, Leclercq et al. [139] reported that corneal kindling with 6 Hz stimuli results in fully kindled male NMRI mice that are markedly less susceptible to ASMs than mice kindled with 50 Hz stimuli. All tested ASMs (clonazepam, valproate, carbamazepine, and levetiracetam) showed a relatively lower potency in the 6 Hz kindling model and a limited efficacy against focal seizures was observed with carbamazepine and levetiracetam. The authors concluded that the observed low potency and limited efficacy of ASMs in 6 Hz fully kindled mice suggest that this model could be a useful tool in the discovery of novel ASMs targeting treatment-resistant epilepsy [139].

### 3.6. The Amygdala-Kindled Mouse Model of TLE

Interestingly, more recently, Leclercq et al. [140] reported that the amygdala-kindled male C57BL/6J mouse is more resistant to ASMs than the amygdala-kindled male Sprague-Dawley rat. Indeed, of the 9 ASMs that were tested in kindled mice, ED_50_s against generalized seizures could only be determined for brivaracetam (68 mg/kg) and valproate (239 mg/kg), whereas most other ASMs only displayed partial protection (Figure 1). The investigational compound, padsevonil, was the most potent substance in this model (ED_50_ 1.2 mg/kg). In line with its high potency in the mouse kindling model, padsevonil was also the most potent compound in the 6 Hz seizure test in mice (ED_50_ 0.16 mg/kg). However, padsevonil failed in two multicenter, randomized, double-blind, placebo-controlled, parallel-group trials in patients with drug-resistant focal-onset epilepsy [141].

### 3.7. Advantages and Limitations of Animal Models with Induced Seizures That Are Resistant to ASMs

The main advantage of these models is that seizures are induced at will, thus these models do not require any continuous EEG video monitoring of seizures. They are, therefore, well suited for the identification and differentiation of novel compounds with presumed anti-seizure effects. Furthermore, because this category of models includes both the induction of seizures in naïve rodents and kindled or epileptic rodents, the impact of epileptogenic brain alterations on drug potencies, efficacies, and adverse effects can be examined. The main objection is that the pharmacology of induced and spontaneous seizures may differ, which is discussed in Section 5. Furthermore, the animal models with induced seizures described in Section 3 are less well suited to study mechanisms of ASM resistance than the models described in Section 2.

## 4. Animal Models with Spontaneous Seizures That Are Resistant to ASMs

Post-SE models of TLE such as the pilocarpine and BLA-SE models have been already discussed in Section 2.5 and Section 2.6, i.e., when describing their use for selecting phenobarbital responsive and resistant rats. More recently, models inducing an SE by kainate either systemically or intracerebrally have been described to lead to SRSs that are resistant to several ASMs (see below). To the best of our knowledge, the pharmacology of SRSs developing in stroke, traumatic brain injury (TBI), or encephalitis models has not been characterized to any extent because, in contrast to post-SE models, only a small percentage of rodents develop convulsive SRSs in such models and it may take several months before SRSs occur [142,143,144]. The group of D’Ambrosio reported that rostral parasagittal fluid percussion injury (rpFPI) in young (32–36 days of age) male Sprague–Dawley rats, in which the initiating rpFPI closely replicates contusive closed-head injury in humans, results in spontaneous focal nonconvulsive seizures in >80% of the rats within 4 weeks following rpFPI [145,146]. These seizures were poorly responsive to subchronic treatment with carbamazepine, valproate, and carisbamate [147,148] but were prevented by mild focal cooling and brivaracetam [149,150]. However, the definition of focal seizures used by D’Ambrosio et al. and their use for drug testing has been criticized [151,152,153], although the principal findings of D’Ambrosio were reproduced more recently by another group [154]. Furthermore, by increasing the severity of lateral FPI-induced brain injury, Smith et al. [155] reported that >60% of rats exhibit convulsive SRSs within 2–5 weeks after FPI, which would make the model more suitable for drug testing, provided these findings can be replicated by other groups.

In some genetic mouse or rat models, seizures either occur spontaneously or can be induced by sensory stimulation such as heat or sound [156,157]. The pharmacology of some of these genetic models will be discussed in Section 4.3.

### 4.1. Post-SE Models of TLE

Among the various available post-SE models of epilepsy, mouse models in which epilepsy is induced by intracerebral injection of the excitotoxic glutamate receptor agonist kainate have become increasingly popular in recent years [158,159]. Compared with rodent models in which SE is induced by systemic administration of kainate or the muscarinic receptor agonist pilocarpine, epilepsy induced by focal unilateral injection of kainate into either hippocampus or amygdala resembles mTLE more closely, particularly because the widespread brain damage observed with systemic convulsants is avoided.

#### 4.1.1. The Intrahippocampal Kainate Mouse and Rat Models of TLE

The first studies on the consequences of IHK in rats and mice were published some 40 years ago [160,161,162,163], but the model became popular much more recently after Bouilleret et al. reported in 1999 that SRSs develop in mice after intrahippocampal injection of kainate [164]. The same group reported in 2004 that the frequent electrographic seizures recorded from the kainate focus in the ipsilateral hippocampus (“hippocampal paroxysmal discharges”; HPDs) were not responsive to acute carbamazepine, phenytoin, or valproate treatment, but could be suppressed by diazepam [165]. This prompted the Löscher lab and other groups to characterize the pharmacology of the HPDs in more detail. Depending on the mouse strain, spontaneous HPDs occur several times per hour, thus allowing them to be used for acute (single-dose) drug testing. Similarly, a second type of electrographic seizures, high-voltage sharp waves (HVSWs), can be recorded from the hippocampus and used for drug testing [25,166], whereas generalized convulsive electroclinical seizures occur only infrequently in this model. For the purpose of drug resistance, we will restrict the discussion to HPDs. HPDs may be associated with subtle focal seizure-like clinical alterations [29], but most groups using this model for drug testing characterize them in the hippocampal EEG only.

The group of Corinne Roucard (SynapCell, La Tronche, France) uses the IHK mouse model of mTLE for the characterization of novel compounds and this assay is performed as a subcontract to *SynapCell* in the ETSP (Figure 7). Figure 9 illustrates the ED_50_s of clinically approved ASMs determined by SynapCell in this model in male C57BL/6J mice [167] in comparison with MES ED_50_s and rotarod TD_50_s in normal mice taken from the literature. As shown in this figure, in the IHK model, phenytoin, carbamazepine, lamotrigine, and valproate are either not effective enough to allow calculation of ED_50_s or ED_50_s are in the same dose range as TD_50_s in the rotarod test or other models for determining “minimal neurotoxicity”. In contrast, diazepam, phenobarbital, vigabatrin, tiagabine, levetiracetam, and pregabalin are quite effective in this model (Figure 9).

Compared with models with systemic administration of kainate or pilocarpine, which induces bilateral and widespread brain damage, one important advantage of the IHK mouse model is that the neuropathology in this model is mostly unilateral and much more restricted, reproducing most of the morphological characteristics of mTLE with hippocampal sclerosis (neuronal loss, gliosis, reorganization of neurotransmitter receptors, mossy fiber sprouting, and granule cell dispersion) observed in patients with this type of epilepsy [29,159]. This is similar in the IHK rat model [91]. However, in contrast to mice, rats do not exhibit the frequent HPDs that are the main advantage of the mouse model for drug testing [168]. The predominant type of SRSs observed in the rat IHK during video EEG monitoring is stage 4/5 generalized convulsive, but also focal (stage 2 or 3) seizures are recorded [169]. Both focal and generalized convulsive seizures are associated with paroxysmal activity in the hippocampal EEG. A drug trial with prolonged daily administration of phenobarbital showed that all epileptic rats used in this trial responded to treatment with suppression of SRSs [169].

#### 4.1.2. The Intra-Amygdala Kainate Mouse Model of TLE

Until recently, the IAK model was used almost exclusively for studies on epileptogenesis and antiepileptogenesis, whereas the IHK model has been used for both ASM and antiepileptogenesis studies [159]. In 2020, the Löscher lab reported a face-to-face comparison of the IAK and IHK mouse models of mTLE and their utility for testing novel therapies [170]. When SRSs were recorded from the ipsilateral hippocampus, relatively frequent electroclinical seizures were determined in the IAK model, whereas only infrequent electroclinical seizures but extremely frequent focal electrographic seizures were determined in the IHK model. As a consequence of the differences in SRS frequency, prolonged video electroencephalographic monitoring and drug administration were needed for testing the efficacy of the benchmark ASM carbamazepine in the intra-amygdala kainate model, whereas acute drug testing was possible in the IHK model. In both models, carbamazepine was only effective at high doses, indicating resistance to this benchmark drug [170].

In a subsequent study by West et al. [171], various ASMs were evaluated for their effects on SRSs in the IAK mouse model: phenytoin (20 mg/kg, b.i.d.), carbamazepine (30 mg/kg, t.i.d.), valproate (240 mg/kg, t.i.d.), diazepam (4 mg/kg, b.i.d.), and phenobarbital (25 and 50 mg/kg, b.i.d.). Valproate, diazepam, and phenobarbital significantly attenuated SRS frequency relative to vehicle controls at doses devoid of observable adverse behavioral effects. However, only diazepam provided significant seizure freedom. Neither phenytoin nor carbamazepine significantly altered SRS frequency or freedom under these experimental conditions. The authors concluded that SRSs in this IAK model of mTLE are pharmacoresistant to two representative sodium channel-inhibiting ASMs (phenytoin and carbamazepine) and partially sensitive to GABA-receptor-modulating ASMs (diazepam and phenobarbital) or a mixed-mechanism ASM (valproate). Accordingly, this model has now been incorporated into the NINDS- ETSP testing platform for treatment-resistant epilepsy (Figure 7).

#### 4.1.3. The Systemic Kainate Rat Model of TLE

To the best of our knowledge, the group of Ed Dudek was the first to use SRS development after SE induction by systemic (i.p.) injection of kainate for systematic drug testing. In this first study, kainate was i.p. administered in repeated low doses (5 mg/kg) every hour until each Sprague–Dawley rat experienced convulsive SE for >3 h [172]. Following the development of spontaneous seizures, 6 1-month trials (*n* = 6–10 rats) assessed the effects of single daily doses of topiramate (0.3–100 mg/kg) on SRSs. Each trial involved six pairs of topiramate and saline-control treatments administered as i.p. injections on alternate days with a recovery day between each treatment day, using a repeated-measures, crossover protocol. Thus, this approach essentially compared each seizure frequency for a particular drug treatment with the seizure frequency of a saline treatment directly before or after the drug treatment in question. A significant effect of topiramate was observed for 12 h (i.e., two 6 h periods) after a 30 mg/kg injection, and full recovery from the drug effect was complete within 43 h. Topiramate exerted a significant effect at doses of 10 mg/kg, 30 mg/kg, and 100 mg/kg, and the effects of topiramate (0.3–100 mg/kg) were dose dependent. Even at the highest doses (30 and 100 mg/kg), topiramate only decreased the relative seizure frequency to 0.44 and did not block all seizures, which may indicate the resistance of SRSs in this model. The authors concluded that the kainate rat model of TLE can be used efficiently for rapid testing of ASMs with a repeated-measures, crossover protocol [172]. However, SRSs were monitored by video only and the short half-life of topiramate in rats (~2.5 h) may have contributed to its relatively poor efficacy.

In a subsequent study, Grabenstatter et al. (2007) used the same protocol for testing carbamazepine. When single i.p. injections of carbamazepine (10–100 mg/kg) were compared with vehicle injections, the ASM significantly reduced motor seizure frequency at 30 mg/kg and 100 mg/kg, and these doses completely blocked motor seizures during a 6 h postdrug epoch in 25% and 70% of the animals, respectively. Single administrations of 30 mg/kg and 100 mg/kg carbamazepine in food also significantly reduced motor seizures, and blocked seizures in 33% and 89% of the rats, respectively. The TD_50_ of carbamazepine in the rotarod test in rats is 37 mg/kg i.p. [43], indicating that the effective doses of carbamazepine in the kainate model were in the neurotoxic range of this drug in rats. When similar protocols were used with carbamazepine-containing or control food pellets, carbamazepine (100 mg/kg) administered in food 3 times per day continuously blocked nearly all motor seizures over 5 days, and completely suppressed motor seizures in 50% of the animals tested [173]. Although no tests of toxicity were conducted, there was no obvious sedation, ataxia, or weight changes observed in animals during continuous carbamazepine treatment relative to control treatment periods.

More recently, Grabenstatter and Dudek [174] used chronic video EEG monitoring to reassess the effects of carbamazepine with a repeated-measures, crossover protocol in the kainate rat model. Carbamazepine-induced decreases in seizure frequency were not significant at 10 mg/kg i.p.; however, at 30 mg/kg, seizure frequency was significantly reduced for convulsive but not nonconvulsive seizures, indicating that focal (nonconvulsive) seizures were more resistant than secondarily generalized convulsive seizures in this model. At 100 mg/kg, CBZ strongly suppressed both convulsive and nonconvulsive seizures. Again, as with topiramate, the rapid elimination of carbamazepine in rats (half-life 1.2–3.5 h) is a problem for protocols using single daily doses of such a drug [93].

Thomson and White [175] developed a novel computer-automated pellet delivery system that allows for tight experimenter control of treatment using a drug-in-food protocol in food-restricted animals. This system was used to study the pharmacokinetics of oral carbamazepine in rats when treated with 75 mg/kg every 6 h for 10 days [176]. Observed and predicted carbamazepine concentrations were maintained within the therapeutic window (4–12 μg/mL) for the duration of the study. In another study with carbamazepine in the kainate rat model, the automated feeder system was used to study medication nonadherence, which directly contributes to poor seizure control [177]. Correction of carbamazepine nonadherence resulted in improved seizure control. The automated medicated pellet delivery system will likely resolve the problems associated with the chronic delivery of drugs with a short half-life in rodents [93], provided that any noted difficulties arising from poor drug palatability can be resolved.

Recently, Thomson et al. [178] evaluated the anti-seizure efficacies of 16 ASMs in the systemic kainate rat model, using a 2-week crossover design and i.p. drug administration at dose intervals (1–3 treatments/d) that were selected based on known half-lives for each compound in rats. Continuous (24/7) video EEG monitoring was used to record the SRSs. None of the 16 ASMs conferred complete seizure freedom during the testing period at the doses tested, indicating high SRS resistance in this model (Figure 1). Carbamazepine (90 mg/kg/d), phenobarbital (30 mg/kg/d), and ezogabine (15 mg/kg/d) significantly reduced seizure burden at the doses evaluated. In addition, a dose-response study of topiramate (20–600 mg/kg/d) demonstrated that this compound reduced seizure burden at both therapeutic and supratherapeutic doses. The authors concluded that this screening paradigm may be useful for testing novel compounds with potential utility in DRE [178] and, consequently, the model was implemented in the NINDS- ETSP (Figure 7).

#### 4.1.4. The Systemic Pilocarpine Rat and Mouse Models of TLE

Surprisingly, only a few studies examined the pharmacological susceptibility of SRSs in the pilocarpine or lithium–pilocarpine models of TLE. Leite and Cavalheiro [179] studied the anti-seizure effect of several ASMs (i.e., phenobarbital (40 mg/kg/d), carbamazepine (120 mg/kg/d), phenytoin (100 mg/kg/d), valproate (450 mg/kd/d and 600 mg/kg/d), and ethosuximide (400 mg/kg/d)) against SRSs in the pilocarpine model in rats and found that, except for ethosuximide, all ASMs effectively blocked the seizures when administered over 2 weeks. However, tolerance to the anti-seizure effect of phenobarbital, carbamazepine, and valproate was observed, in that SRSs were more effectively blocked in the first week of treatment compared with the second week. Furthermore, at the high doses of ASMs administered, marked adverse effects, such as sedation and muscle relaxation, were observed. A significant limitation of the study of Leite and Cavalheiro (1995) is that SRSs were only behaviorally monitored (by direct observation without EEG) and only for 10 h/day (8. a.m.–6 p.m.), 5 days/week. Thus, seizures occurring at night and subtle focal seizures were not recorded.

Studies of the Löscher lab with levetiracetam [103] and phenobarbital [29] in the lithium–pilocarpine model are described in Section 2.6. As described in this section, by using continuous video EEG monitoring of SRSs before, during, and after prolonged treatment with phenobarbital at maximum tolerated doses (loading dose 25 mg/kg i.p., followed by 15 mg/kg b.i.d.), SRSs differed in their response in individual rats [23], and this was also observed with levetiracetam [103].

Chakir et al. [180] examined the effect of delayed treatment with an ASM in the pilocarpine rat model. Animals with SRSs were treated for 50 days with carbamazepine (50 mg/kg i.p., 3 times daily), starting either at the onset of SRSs (27 days after SE) or 50 days later and compared with epileptic untreated rats. Prompt administration of carbamazepine suppressed convulsive seizures, whereas delayed treatment only decreased their frequency, indicating drug resistance.

Systemic injection of pilocarpine is also used for induction of SE and subsequent SRSs in mice [89]. The study of Moon et al. [104], in which ASM responders and nonresponders were selected by levetiracetam and valproate, is described in Section 2.6.

#### 4.1.5. Advantages and Limitations of Animal Models of TLE with Spontaneous Seizures That Are Resistant to ASMs

As shown in Figure 7, the chronic epilepsy models described in this section are useful for the differentiation of novel anti-seizure compounds. These models allow studying whether drug potencies and efficacies differ from those determined in models with induced seizures (see Section 5). Furthermore, because most models with SRSs necessitate prolonged drug administration, this allows drug experiments examining whether drug effects change during the trial phase, e.g., because of the development of tolerance [131]. However, because inter-individual differences in drug effects are typically not examined when using these models during the differentiation phase illustrated in Figure 7, important information may be missed (see Section 2).

### 4.2. Models of Focal Neocortical Epilepsy

As with TLE, focal neocortical epilepsy is often drug-resistant [181]. There are multiple causes of neocortical epilepsy, including head trauma, stroke, brain tumors, brain infections, and cortical dysplasia [182]. Head trauma causes neocortical epilepsy more often than other forms of focal epilepsy (e.g., mTLE). The clinical manifestations of neocortical seizures depend on the localization of seizure onset and initial seizure propagation. A variety of animal models of focal neocortical epilepsy have been described but the majority of these models have not been pharmacologically characterized to any significant extent [183,184,185,186]. Prominent animal models of focal neocortical epilepsy with SRSs include the focal penicillin model in rats or cats, cortically implanted metals such as aluminum hydroxide or cobalt chloride in cats or monkeys, cortical undercut in cats, rats, and mice, or the tetanus toxin model in cats or rats. Currently, the latter model is among the most widely used models of focal neocortical epilepsy [187]. When tetanus toxin is injected into the motor neocortex of male Sprague-Dawley rats, the spontaneous focal seizures are refractory to diazepam and phenytoin [188]. The latter group used the model for evaluating novel anti-seizure therapies [188,189,190], but another group of researchers failed to reproduce stable epileptic activity in this model [191]. Therefore, Rassner et al. [191] combined focal neocortical injection of tetanus toxin with cobalt chloride that was placed topically on the neocortical surface. Although no clear seizures were observed, the rats exhibited epileptiform cortical potentials, which could be reduced by local valproate-containing polymer implants that were implanted above the neocortical focus.

### 4.3. Genetic Models

Genetic animal models of epilepsy comprise genetically predisposed animal species in which seizures either occur spontaneously or in response to sensory stimulation. Rodent models with reflex seizures comprise audiogenic seizure susceptible mice (e.g., the DBA/2 mouse) and rats (e.g., the Genetically Epilepsy-Prone Rat (GEPR)), and gerbils with seizures in response to different sensory stimuli [156,157,192]. In general, seizures in these models are susceptible to a wide range of ASMs. Genetic animal models with spontaneous seizures include naturally occurring (often serendipitously discovered) epileptic rodents (e.g., the *tottering* mouse or the Genetic Absence Epilepsy Rat of Strasbourg (GAERS)), which can be monogenic (with a known mutation) or polygenic, and gene-targeted models carrying an engineered mutation in their genome causing epileptic seizures [156,193,194,195].

More recently, through advances in genetic sequencing technologies, a vast number of genes have been implicated in developmental and epileptic encephalopathies [196]. State-of-the-art gene-editing techniques have led to the generation of hundreds of mouse models of these rare but devastating and largely intractable childhood epilepsies [197]. However, despite the specific potential of genetic animal models of epilepsy for the development of new, highly specific ASMs, most of them are only rarely used in preclinical evaluation of ASMs to date. One important exception are mouse models of Dravet syndrome (DS), the most common epileptic encephalopathy [198], and genetically engineered zebrafish [199].

More than 80% of patients with DS have a loss-of-function mutation in the *SCN1A* gene that encodes the alpha subunit of voltage-gated sodium channel Na_v_1.1 [200]. Many first-line ASMs are ineffective in treating seizures in DS patients, and sodium channel modulators such as carbamazepine are known to exacerbate seizures (Table 1). Retrospective studies of ASM responses of DS patients rank benzodiazepines (clobazam, clonazepam), valproate, or stiripentol as the most effective options and these ASMs are considered the standard of care “first line” treatment for most of this patient population [198,201]. Furthermore, recent clinical trials show positive seizure reduction utilizing cannabidiol and, more effectively, fenfluramine [202]. However, most DS patients need polytherapy with these ASMs and even then many patients do not respond adequately [198].

**Table 1 cells-12-01233-t001:** Predictive validity of animal models of Dravet syndrome. The table was modified from Griffin et al. [198]. Data are from Oakley et al. [203], Hawkins et al. [204], Kaplan et al. [205], and Thornton et al. [206]. “?” indicates that no data were found.

ASM	Anti-Seizure Activity			
	DS Patients (Spontaneous Seizures; Chronic Treatment) *	*Scn1a*^+/−^ Mice		*Scn1lab^s552^* Zebrafish (Spontaneous Seizures; Acute Treatment)
		Spontaneous Seizures (Subchronic Treatment)	Hyperthermia- Induced Seizures (Single-Dose Treatment)	
Valproate	Yes	No (no **)	Yes ***	Yes
Clobazam	Yes	No (yes **)	Yes	Yes
Stiripentol	Yes	No (no **)	Yes ***	Yes
Topiramate	Yes	No	No	Yes
Clonazepam	Yes	?	Yes	Yes
Fenfluramine	Yes	?	?	Yes
Cannabidiol	Yes	Yes	Yes	Yes
Stiripentol plus clobazam	Yes	Yes **	Yes	?
Levetiracetam	No	No	Yes	No
Phenobarbital	No	No	Yes	No
Lamotrigine	No (worse)	No (worse)	No (worse)	No
Phenytoin	No (worse)	No	No	No
Carbamazepine	No (worse)	No	No	No (worse)
No. of ASMs predictive		8/11	9/11	12/12

* Note that most DS patients need polytherapy with these ASMs and even then many patients do not adequately respond. ** After hyperthermic priming of spontaneous seizures. *** Effect only at doses resulting in plasma levels that exceed the human therapeutic range

Currently, there are numerous genetic mouse models for DS, which aim to replicate the *SCN1A* loss-of-function observed in DS [197,198]. These models display infrequent spontaneous ictal and interictal EEG activity and seizures, as well as myoclonic jerks and flexions. Seizures in *Scn1a* knock-out mice are also inducible via hyperthermia [198]. However, simply inducing seizures in an animal with a Na_v_1.1 deficiency comes with the risk of identifying compounds that impact the seizure-induction mechanism and will ultimately not be effective against spontaneous seizures in DS patients [198].

Based on clinical response data in DS patients, Hawkins et al. (2017) chose nine ASMs and evaluated them in *Scn1a*^+/−^ mice for effects on hyperthermia-induced seizures vs. spontaneous generalized tonic–clonic seizures (GTCS). Data are summarized in Table 1. Only for 4/9 ASMs tested, the same type of response was found on hyperthermia-induced vs. spontaneous seizures. Compared with the clinical effects of the nine ASMs tested by Hawkins et al. [204], ASM effects on hyperthermia-induced seizures in DS mice were predictive for 7/9 ASMs, whereas ASM testing on SRSs only correctly predicted the lack of efficacy of some ASMs in DS patients. Thus, the mouse model failed to show efficacy in suppressing spontaneous seizures for any “first-line” DS drugs (but see the ASM combination experiments below). Hawkins et al. [204] suggested that the lack of a significant response of first-line DS drugs in the mouse model may have been due to a floor effect, as the untreated *Scn1a*^+/−^ mice had relatively low spontaneous GTCS incidence and frequencies, so the model was underpowered to identify compounds that significantly reduced spontaneous GTCS frequency within the observation window. Therefore, they modified the model by using hyperthermic priming of SRSs to increase SRS frequency and re-examined three ASMs, clobazam, stiripentol, and valproate. Only clobazam (but not stiripentol or valproate) exhibited significant anti-seizure efficacy in this modified mouse model (Table 1). Furthermore, cannabidiol was shown to suppress SRSs in the DS mouse model [205]), which is in line with the (limited) clinical efficacy of this drug in DS patients (Table 1).

Since most DS patients are treated with ASM combinations, for instance, stiripentol and clobazam, Hawkins et al. [204] also tested combinations of stiripentol and clobazam on both hyperthermia-induced and spontaneous seizures in the DS mouse model. Significant anti-seizure effects were obtained with both types of seizures (Table 1).

More recently, the DS mouse model was evaluated as a drug screening platform by the ETSP program of the NINDS/NIH, using the conditional knock-in *Scn1a^A1783V/WT^* mouse [207]. ASMs were considered effective if they significantly increased the temperature at which the DS mice had seizures. The data showed that hyperthermia-induced seizures in this model of DS are highly refractory to a battery of ASMs. Exceptions were clobazam, tiagabine, levetiracetam, and the combination of clobazam and valproate with add-on stiripentol, which elevated seizure thresholds.

Zebrafish larvae offer an alternate in vivo model for assessing the anti-seizure efficacy of ASMs for DS [198]. Using scn1lab^s552^ mutant zebrafish with unprovoked seizure activity, a variety of ASMs have been evaluated, showing a high predictive validity of this model (Table 1). Because of the rapidity with which known human mutations can be expressed in zebrafish, the zebrafish model is now widely used in the search for novel, more effective ASMs for different genetic epilepsies, including DS [198,199,208,209]. Indeed, zebrafish is a vertebrate genetic model organism with tremendous potential for modeling genetic epilepsies and for the rapid screening of investigational drugs.

However, as with other animal models, there are limitations and drawbacks in the zebrafish model [17,198,210]. The main one is inherent to its ability as a model of drug-resistant epilepsies: the potential of zebrafish to identify novel more effective drugs remains to be proven. In this respect, the outcome of ongoing clinical trials with clemizole as an add-on therapy to control drug-resistant convulsive seizures in patients with DS will be important because this antihistamine was identified as a potential novel treatment of this syndrome by a phenotypic screening of drug libraries in zebrafish *scn1* mutants [211]. A subsequent study showed that clemizole binds to serotonin receptors and its anti-seizure activity in zebrafish can be mimicked by drugs acting on serotonin signaling pathways [212].

Another limitation of the zebrafish model is the lack of pharmacokinetic control when administering drugs via the fish tank water. With water-soluble drugs, exposure to the drug remains constant as the larvae are immersed in bathing media containing the drug, which is rapidly absorbed through the skin and gills. However, quantifying drug uptake into zebrafish larvae remains a limitation of this model, and differences in drug absorption are unavoidable. Thus, it is not clear how the effective concentrations in larvae can be related to appropriate effect levels in mammalian models. Furthermore, highly hydrophobic compounds, large molecules, and proteins are not absorbed from the water, but rather need to be injected into the animals, which limits large-scale testing. Finally, the zebrafish model does not replace any rodent models described here but provides a unique platform for targeting genetic mutations involved in epilepsy.

## 5. Pharmacology of Induced vs. Spontaneous Seizures in Animal Models of Drug-Resistant Epilepsy

A comparison of the pharmacology of chronic epilepsy models with models of acute (induced) seizures in previously healthy (non-epileptic) animals, such as the MES test, demonstrates that drug testing in chronic models of epilepsy yields data that are more predictive of clinical efficacy and adverse effects; so chronic models should be used relatively early in drug development to minimize false positives [88]. Interestingly, the pharmacology of elicited kindled seizures in fully kindled rats and SRSs in post-SE models of TLE is remarkably similar [88], which is also reported here.

The lower part of Figure 1 summarizes the effects of various ASMs in nine rodent models which are per se resistant to some ASMs. In five of these models, seizures are induced while in three models seizures occur spontaneously. In all nine models, seizures are resistant to sodium channel modulators such as phenytoin and lamotrigine, whereas GABA-potentiating ASMs, such as phenobarbital and benzodiazepines are quite effective in most models. The reason for this discrepancy is not clear. Importantly, it does not reflect the clinical reality because patients with drug-resistant seizures are typically resistant to different mechanistic categories of ASMs, including those that act at least in part by potentiating GABAergic transmission [13,14,213]. Apart from ASMs that act by sodium channel modulation or GABA potentiation, differences occur across the nine models of drug-resistant seizures illustrated in Figure 1. For instance, lamotrigine-resistant 60 Hz corneal-kindled mice are resistant to retigabine while this drug is effective in most other models. In apparent contrast to the high anti-seizure efficacy of GABA-potentiating drugs in animal models that are per se resistant to sodium channel modulators and some other ASMs, such a difference is not observed in the phenytoin-resistant amygdala-kindled rat (Figure 1 and Figure 2B). Thus, the latter model appears to reflect the clinical reality better than any of the other models illustrated in Figure 1. This can be explained by the many molecular alterations that have been found in phenytoin-resistant kindled rats compared with phenytoin responders in this model (see Section 2.3.2 and Section 2.3.3).

Overall, this strongly indicates that a battery of animal models of drug-resistant seizures rather than a single test should be used during the identification and differentiation of investigational compounds. This strategy is nicely illustrated by the ETSP (Figure 7), which incorporates several of the models shown in Figure 1 in its DRE workflow.

## 6. Evaluation of Drug Combinations vs. Single Drug Testing

Patients with DRE are typically treated with more than one ASM. Robust evidence to guide clinicians on when and how to combine ASMs is lacking, and current practice recommendations are largely empirical [214,215,216,217]. A popular strategy for combination therapy is a pharmacomechanistic approach based on the (perceived) modes of action of ASMs. For instance, Deckers et al. (2000) reviewed the available animal and human data and concluded that combinations involving a sodium channel modulator and a drug with GABAergic properties appeared to be particularly beneficial. Indeed, one of the few clinically proven synergistic ASM combinations is a combination of lamotrigine and valproate [216,218].

The principle underlying the pharmacomechanistic approach (“rational polypharmacy”) is that the combination of two medications with differing mechanisms of action may result in supra-additive or synergistic anti-seizure effects, with infra-additive toxicity [216]. The inherent problem with the concept of rational polypharmacy is that the number of clinically approved ASMs is so high that several 100 dual therapies and more than 1000 triple combinations are possible, making any systematic evidence-based clinical evaluation impossible. This issue becomes even more complex when considering the multiple possible dose ratios between two or more ASMs in combination. Therefore, a systematic evaluation of ASM combinations is only feasible in animal models [216,219,220]. Hundreds of different ASM combinations have been evaluated preclinically [216,219,220,221,222], but the 6 Hz mouse model is the only animal model of drug-resistant seizures for which some systematic data on ASM combinations are available [223]. Luszczki et al. [223] determined possible interactions among 5 second-generation ASMs, i.e., gabapentin, lacosamide, levetiracetam, pregabalin, and retigabine in the 6 Hz corneal stimulation-induced seizure model in adult male albino Swiss mice. The anti-seizure effects of 10 various two-drug combinations of these ASMs were evaluated with type I isobolographic analysis associated with the graphical presentation of a polygonogram to visualize the types of interactions. The most beneficial combination, offering the highest level of synergistic suppression of seizures in mice was that of levetiracetam and retigabine. In a previous study by the same group, interactions of levetiracetam with various ASMs (clonazepam, oxcarbazepine, phenobarbital, tiagabine, and valproate) were examined in the 6 Hz mouse model [224]. Using isobolographic analysis, a combination of levetiracetam and phenobarbital exerted supra-additive (synergistic) anti-seizure activity.

## 7. Evaluation of Drug Potency vs. Efficacy

The typical approach of ASM testing in animal models primarily focuses on drug potency and not efficacy [3]. Thus, different investigational drugs are compared in terms of their anti-seizure ED_50_s, i.e., the dose-suppressing seizures in 50% of the animals, which is calculated from dose-response curves, testing one group of animals per dose. The lower the ED_50_, the more potent the drug, and high potency is often an important argument for selecting drugs for further development. However, it is the anti-seizure efficacy that finally determines the clinical usefulness of a new ASM and this should be considered during preclinical drug testing [17].

The concepts of potency and efficacy are often confused and used interchangeably within the scientific community and the pharmaceutical industry [225]. Whereas drug potency is the amount of drug that is needed to produce a defined effect (e.g., seizure suppression in 50% of the animals), drug efficacy is the maximum effect that a drug can produce (see Figure 10A). Clinical efficacy measures the magnitude and profile of clinical disease improvement after drug administration. Drug efficacy is principally determined by the interaction between the drug and its receptor–effector system. For instance, a full agonist at a given receptor has higher efficacy than a partial agonist although both compounds may have similar potencies in an animal model [226]. Although preclinical drug potency can be a good preclinical marker of the therapeutic potential of a drug, it does not predict clinical efficacy. One recent example is the clinical efficacy of padsevonil and cenobamate, which are both highly potent in diverse animal models of seizures and epilepsy [116,216]; however, while cenobamate proved to be highly effective in several clinical trials in patients with drug-resistant focal epilepsy, padsevonil failed in such trials [216].

In animal models of drug-resistant seizures, efficacy is more difficult to determine than potency, but one approach is determining ED_50_s in the 6 Hz model at increasing current intensities (22 mA, 32 mA, and 44 mA) as proposed by Barton et al. (2001). With increasing currents, most ASMs become less effective or ineffective in this test (Figure 10B), but a few third-generation ASMs (e.g., cenobamate) remain effective [116], thus allowing the differentiation of ASMs in terms of anti-seizure efficacy in this mouse model. Interestingly, the novel ASM cenobamate is also quite effective in patients with difficult-to-treat focal epilepsies [117].

Another approach has been illustrated by testing a large series of ASMs in phenytoin-nonresponding and phenytoin-responding kindled rats (see Section 2.3.1). Interestingly, in these experiments levetiracetam was the only ASM that was highly effective in phenytoin-resistant rats, whereas all other ASMs were significantly less efficacious or not efficacious at all in phenytoin nonresponders, demonstrating that the phenytoin resistance in this model extends to various other ASMs (Figure 4A), thus simulating multidrug resistance in patients with TLE. One might expect that, based on these preclinical findings, levetiracetam could possibly be more effective than other ASMs in the treatment of drug-resistant focal epilepsy. Indeed, levetiracetam has become a benchmark in the clinical management of focal epilepsy [217]. However, randomized controlled clinical trials with face-to-face comparisons of ASMs in such patients are hardly available.

## 8. The Importance of Pharmacokinetics for Anti-Seizure Efficacy Testing in Animal Models

Preclinical pharmacokinetic/pharmacodynamic modeling and simulation are essential parts of any drug development in the pharmaceutical industry [227]. The impact of pharmacokinetics in rodents for critically affecting the anti-seizure efficacy of ASMs and investigational compounds has been highlighted throughout this review. As shown in Table 2, most ASMs are much more rapidly eliminated by rats and mice than humans and the same is true for investigational compounds. This is typically a result of the higher metabolic rate of rodents compared with humans [93,228,229]. This also affects dose conversion between rodents and humans, so allometric scaling based on body surface area, which is related to the metabolic rate of an animal and is established through the evolutionary adaptation of animals to their size, is widely used in such a dose conversion [229]. For calculating human equivalent doses in mg/kg bodyweight, scaling factors are 12.3 for the mouse and 6.2 for the rat. Interestingly, when effective plasma concentrations (EC) at anti-seizure ED_50_s of ASMs in seizure models in mice and rats are determined, EC_50_s (in µg/mL) of many ASMs are in the therapeutic plasma concentration range of these ASMs during anti-seizure therapy in epilepsy patients [17,93]. The same is true for drug plasma levels during chronic administration of effective doses of ASMs in rodent models of epilepsy. To the best of our knowledge, Brodie and Reid [230] were the first to propose that, although the dose of drug necessary to produce the same effect may differ considerably between species, the plasma concentration is usually quite similar. This proposal became known as the “Brodie-Reid-Hypothesis” [231]. Thus, plasma levels determined at the time of anti-seizure effect in rodent models can be used for selecting adequate doses of a new ASM for first clinical trials by calculating the doses that will produce such plasma levels in humans [3,17].

In addition to drug elimination, pharmacokinetic analyses in animal experiments allow one to control for whether a drug is sufficiently absorbed from injection sites, reaches the target organ—in the case of ASMs the brain—and whether active metabolites are involved in its action [17]. However, pharmacokinetic principles are only rarely applied by academic scientists in studies on novel drugs or targets. As long as pharmacokinetic experiments are not performed, it is not possible to compare data between models and laboratories. For instance, differences in drug vehicles, the use of drug suspensions instead of solutions, the route of administration, and many other variables can lead to marked differences in a drug’s absorption and distribution and thus critically contribute to the lack of reproducibility of data [93].

Pharmacokinetic analyses are also important to determine the time of peak drug levels (T_max_) following administration. For instance, following i.p. administration of valproate in rodents, peak drug levels in plasma and brain are reached within 10–15 min which also is the time of peak drug effect of this drug in rodent seizure models [232]. Thereafter, the drug is rapidly eliminated, so determining valproate’s anti-seizure effect at, for instance, 30 or 60 min after administration (as often done) would lead to a significant underestimation of its potency.

Diazepam is another example of the marked effect of pharmacokinetics on drug effects in rodent models. In both rodents and humans, diazepam is metabolized to several active metabolites, including desmethyldiazepam (nordiazepam), oxazepam, and temazepam. In rodents, the metabolism of diazepam is so rapid (see Table 2) that it is difficult, if not impossible, to determine the anticonvulsant potency of the parent drug, e.g., in the conventional s.c. PTZ seizure test [233,234]. Using an i.v. PTZ seizure test in which an anti-seizure effect can be determined in mice within 30 sec, diazepam exerted its maximum potency at 1 min after i.v. injection when only traces of desmethyldiazepam could be detected in plasma and the brain [233]. During the next 30 min, the drug was rapidly transformed to desmethyldiazepam and oxazepam, and its anticonvulsant ED_50_ rose by a factor of about 4. At 30 min, plasma and brain levels of desmethyldiazepam were markedly higher than those of the parent drug, so a determination of anticonvulsant potency of diazepam 30 min after i.v. administration would mainly reflect the activity of the metabolite. This is even more marked with i.p. or oral administration of diazepam because first-pass metabolism in the liver rapidly metabolizes most of the parent drug to desmethyldiazepam. In contrast, because the metabolism of diazepam is much slower in humans (Table 2), its onset of action (1–5 min) and duration of peak pharmacological effects (~20–30 min) after i.v. administration in patients (e.g., for treatment of SE) reflects the anti-seizure activity of the parent drug and not its metabolites. The short duration of action of i.v. diazepam in the treatment of SE is mainly due to the rapid and extensive redistribution of this lipid drug from the brain to peripheral tissues and not a result of drug metabolism [235].

Another interesting species difference of diazepam was observed in mice and rats in that the anti-seizure activity following i.v. administered diazepam in the s.c. PTZ seizure test was longer lasting in mice than in rats due to the accumulation of desmethyldiazepam in the mouse but not rat brain [236]. This is a result of the longer half-life of this major diazepam metabolite in mice vs. rats (Table 2).

Concerning the ETSP, information on pharmacokinetics, if available, guide the planning of testing (i.e., dose-selection) of investigational compounds, thereby improving the efficiency, safety, tolerability, and interpretability of the test results generated [112]. However, pharmacokinetic data are not always available, so the program has started to generate such data in house, starting with the reference ASMs levetiracetam, carbamazepine, valproate, and clobazam [237]. Similar to the diazepam data discussed above, distinct pharmacokinetic profiles of ASMs between mice and rats were found. Furthermore, again similar to diazepam, active metabolites contributed to the efficacy of clobazam and carbamazepine [237].

## 9. The Use of Animal Models as Tools for Developing Novel Non-Pharmacological Treatment Strategies

While the focus of this review is the characterization of drug effects in animal models of DRE, such models are also useful for evaluating non-pharmacological strategies for epilepsy therapy. Such strategies include different types of neurostimulation, surgical approaches, dietary treatments, gene therapy, antisense oligonucleotide therapy (ASO), and neurotransplantation, some of which may even cure epilepsy [238,239,240,241,242]. A recent fascinating example of how animal models can be used to develop such strategies is a genetic closed-loop feedback system in mice that was designed to inhibit neurons that participate in seizure activity [243]. To do this, Qiu et al. [243] developed a genetic strategy based on the *Fos* gene, whose expression is up-regulated by neuronal activity, including seizures. The activity-sensitive promoter region of *Fos* was used to drive the expression of a gene encoding an inhibitory protein, voltage-gated potassium channel subunit (Kv1.1, which is encoded by *Kcna1*), which upon channel opening permits efflux of potassium ions and hyperpolarization of the membrane, which reduces action potential generation and neurotransmitter release by the neuron. An adeno-associated virus (AAV) vector encoding the *Fos* promoter and *Kcna1* was used to transfect neurons in vitro and in vivo. In the in vivo experiments, the AAV vector was injected into the hippocampus of mice. *Fos*-driven expression of Kv1.1 in the hippocampus reduced the severity of seizures induced by PTZ and the number of SRSs in the intrahippocampal kainate model. In summary, the results of Qiu et al. [243] describe a sophisticated, localizable new strategy for closed-loop feedback to control seizures [244]. Qiu et al. [243] also showed that the *FOS* promoter-*KCNA1* feedback system is operative in human neurons.

Another interesting approach to treating seizures is the development of optogenetic and chemogenetic therapies for epilepsy [245]. Both types of novel therapies are titratable to suppress seizures without adverse effects. Optogenetics, which uses light to control the excitability of specific neuronal populations, can be used in a closed-loop treatment paradigm, while chemogenetics makes use of endogenous proteins that respond to an exogenous ligand, thereby facilitating translation.

Concerning novel types of neurostimulation, it is interesting to note that dogs with naturally occurring epilepsy are increasingly used as a large animal model [246]. Indeed, other than mice or rats, dogs are large enough to accommodate intracranial EEG and responsive neurostimulation devices designed for humans. In addition to using dogs with naturally occurring epilepsy, several techniques for inducing seizures in laboratory dogs are available [246]. Importantly, the development of vagus nerve stimulation as a novel therapy for DRE in people was based on a series of studies in dogs with induced seizures [246]. Overall, dogs with naturally occurring or induced seizures provide excellent large-animal models to bridge the translational gap between rodents and humans in the development of novel therapies.

## 10. Conclusions

The two categories of rodent models of drug-resistant seizures/epilepsy discussed here (Figure 1) offer different opportunities for basic research and preclinical drug development. Models allowing the selection of ASM responders and nonresponders can be used to decipher the cellular and molecular mechanisms of pharmacoresistance by directly comparing brain alterations in responders vs. nonresponders. Both phenytoin-resistant amygdala-kindled rats and ASM-resistant subgroups of epileptic rats and mice selected from different post-SE models of TLE have been extensively used in this regard. However, these models are too laborious to use for the identification of novel compounds unless the compound is rationally developed to overcome a specific mechanism of drug resistance. For drug screening, rodent models in which seizures can be induced or that exhibit SRSs that occur with high frequency and short latency and are per se resistant to some or various ASMs are very useful. Among the various models shown in Figure 1, the 6 Hz mouse model of drug-resistant focal seizures is currently the most widely used model in preclinical ASM development, but several other models are useful for both the identification and differentiation of investigational compounds.

While each model presents its own strengths and limitations, in general, animal models of drug-resistant seizures demonstrate significant utility in differentiating between investigational ASMs in a preclinical setting. A common strength of many of these models is the ability to reproduce the pathological, behavioral, and pharmacological characteristics of drug-resistant seizures in humans. However, the major limitation arises when it comes to predictive clinical validity or the ability of these models to predict the clinical efficacy of a novel ASM. In other words, no single ASM has made it to the market based on its efficacy solely in any (or all) of these preclinical drug-resistant seizure models. Historically, this may in part be due to the difficulties in conducting this type of research, e.g., laborious, time-consuming, and expensive experiments that result in low-yield/low-throughput metrics. However, with the introduction of newer screening models (e.g., lamotrigine-resistant CKM, mTLE mouse), and the modification of older screening platforms (e.g., the ETSP’s condensed screening paradigm in the post-kainate rat), it may be worth revisiting the characterization of prototype ASM’s to further validate their clinical utility. These types of studies would provide the opportunity to answer important questions about whether we could have predicted the success of certain drugs based on their efficacy in these preclinical animal models. For instance, could the remarkable clinical efficacy of cenobamate in patients with highly refractory focal epilepsy have been predicted by its findings in animal models of drug-resistant seizures, e.g., the simple 6 Hz seizure test?

Positive results from two global, randomized, double-blind, placebo-controlled studies, and a large global, multicenter, open-label safety and pharmacokinetic study with cenobamate in patients with highly refractory focal epilepsy (patients were taking 1–3 concomitant ASMs) demonstrated statistically significant reductions in seizure frequency across all seizure types studied, including focal aware motor, focal impaired awareness, and focal to bilateral tonic–clonic seizures. These clinical trials demonstrated improved seizure control in adults with uncontrolled focal onset seizures taking one to three concomitant ASMs [247,248]. In addition to improved seizure control, cenobamate provided significantly higher seizure freedom rates compared with those reported for patients randomized to receive a placebo on top of their concomitant ASMs [249]. In fact, the degree of seizure freedom (e.g., 20%) associated with cenobamate treatment was well above that seen for any new ASM brought to the market since 1993 when felbamate was approved for the treatment of focal onset seizures for review and discussion see [116,247,248]. As discussed in Guignet et al. [116], cenobamate possesses a broad-spectrum anti-seizure profile in multiple rodent seizure and epilepsy models including the MES, sc PTZ, kindled rat, and GAERS. What separates cenobamate’s preclinical profile from that of other broad-spectrum ASMs is its efficacy in the 6 Hz seizure test (Figure 10B). As tempting as it is to speculate that cenobamate’s sustained efficacy and potency in the 6 Hz test regardless of the stimulus intensity would have predicted the marked efficacy seen in clinical trials, it would be premature to do so and additional study of the utility of the model is certainly warranted.

Additionally, while cenobamate’s marked efficacy in the 6 Hz test is remarkable, it has yet to be studied in any other therapy-resistant seizure model discussed herein. Further investigation into a drug such as cenobamate would be imperative for detailing a picture of the clinical validity of these animal models. Importantly, comparing cenobamate’s preclinical efficacy alongside other ASMs such as levetiracetam, a drug that is very effective in patients with refractory focal epilepsy [250,251] but does not demonstrate the same level of seizure freedom that is seen with cenobamate, would prove valuable in a population of animals that are resistant to other ASMs. Specifically, a major strength of these models is the ability to reproduce the clinical presentation where ~30–40% of patients are refractory to currently available ASMs, a number that has remained stagnant for the last few decades [252]. By identifying and selecting animals that do not respond to two ASMs, i.e., the textbook definition of drug resistance, these models provide the ability to screen for more effective therapies for a difficult-to-treat population. Designing preclinical studies that model human clinical trials in patients with refractory epilepsy would provide valuable insight into the clinical utility of these models and may ultimately change the landscape for predicting the clinical success of new ASMs in reducing the percentage of patients with refractory epilepsy.

While no single animal model can reproduce all the features of the human condition, using a battery of drug-resistant animal models captures many of the behavioral, electroencephalographic, and pharmacological phenomena of drug-resistant seizures. The findings summarized within this review support the use of these models to potentially aid in the development of novel ASMs for treating patients that are not controlled by currently available drugs. Finally, TLE and seizure dynamics are complex, i.e., there are multiple ways in which a seizure can be initiated in the brain, however, combining the use of multiple preclinical models of DRE will hopefully identify better therapies.

The example set by cenobamate’s level of seizure freedom demonstrates that it is possible to find highly effective ASMs using animal seizure and epilepsy models. Whether cenobamate will change the landscape of drug resistance in the future remains to be determined; only time will tell. What is clear is that cenobamate appears to offer something different in seizure control beyond what we have seen over 30 years and, in this respect, the “old models” did give the patient with therapy-resistant epilepsy a novel more efficacious drug and renewed hope, which supports the continued approach to ASM discovery.

## Figures and Tables

**Figure 1 cells-12-01233-f001:**
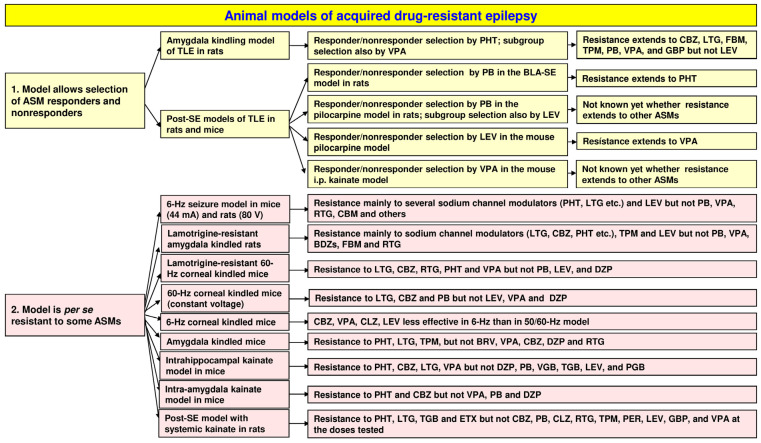
Different categories of mouse and rat models of drug-resistant seizures. See text for details. Abbreviations: ASM, anti-seizure medication; BDZ, benzodiazepine; BRV, brivaracetam; CBM, cenobamate; CBZ, carbamazepine; CLZ, clonazepam; DZP, diazepam; FBM, felbamate; GBP, gabapentin; LCM, lacosamide; LEV, levetiracetam; LTG, lamotrigine; PB, phenobarbital; PER, perampanel; PGB, pregabalin; PHT, phenytoin; RTG, retigabine; SE, status epilepticus; TGB, tiagabine; TLE, temporal lobe epilepsy; TPM, topiramate; VGB, vigabatrin; and VPA, valproate.

**Figure 3 cells-12-01233-f003:**
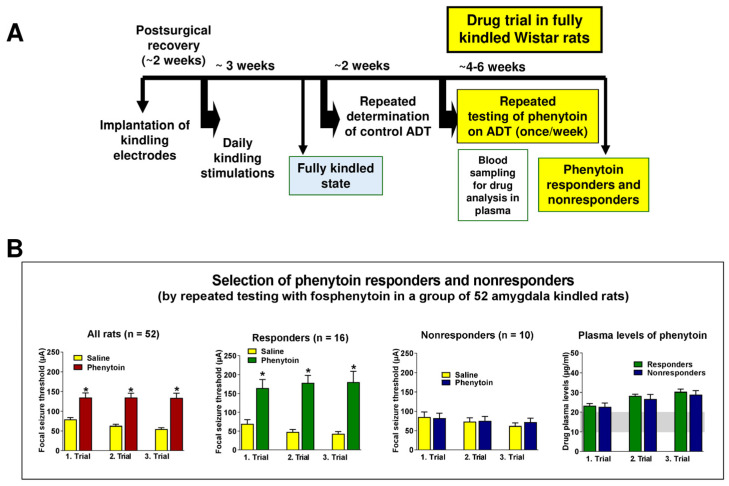
The phenytoin-resistant amygdala-kindled rat model. (**A**) Schematic illustration of selection of drug-refractory (nonresponders) and drug-responsive (responders) rats from the kindling model by repeated testing of female Wistar rats with maximum tolerated doses of phenytoin (75 mg/kg i.p.) or its prodrug fosphenytoin (83.5 mg/kg i.p.). (**B**) Example of a selection of drug-refractory (nonresponders) and drug-responsive (responders) rats from the kindling model by repeated testing with the phenytoin prodrug fosphenytoin. A total of 52 fully-kindled, female Wistar rats were used in this experiment. The anti-seizure effect of a maximum tolerated dose of fosphenytoin (83.5 mg/kg i.p.) was tested by single-dose administration once per week by determining the focal seizure threshold (ADT) in each rat 1 h after drug injection. Control thresholds were determined in each rat 2–3 days before each drug trial, using i.p. injection of saline. All data are shown as means ± S.E.M. Significant differences to control threshold are indicated by asterisks (*p* < 0.001). After each drug injection, blood was sampled for drug analysis in plasma. Only drug trials in which phenytoin plasma levels were within or above the “therapeutic range” in patients with epilepsy (about 10–20 µg/mL) were used for the final evaluation of data. Of the 52 fully kindled rats used in the experiment, 16 rats always responded with a significant ADT increase (responders), whereas 10 rats never showed such an anti-seizure effect (nonresponders). The remaining 26 rats exhibited variable responses to phenytoin (not illustrated). Plasma levels of phenytoin were not different between responders or nonresponders (the shaded area indicates the therapeutic plasma concentration range of phenytoin). Data are from Löscher et al. [17].

**Figure 4 cells-12-01233-f004:**
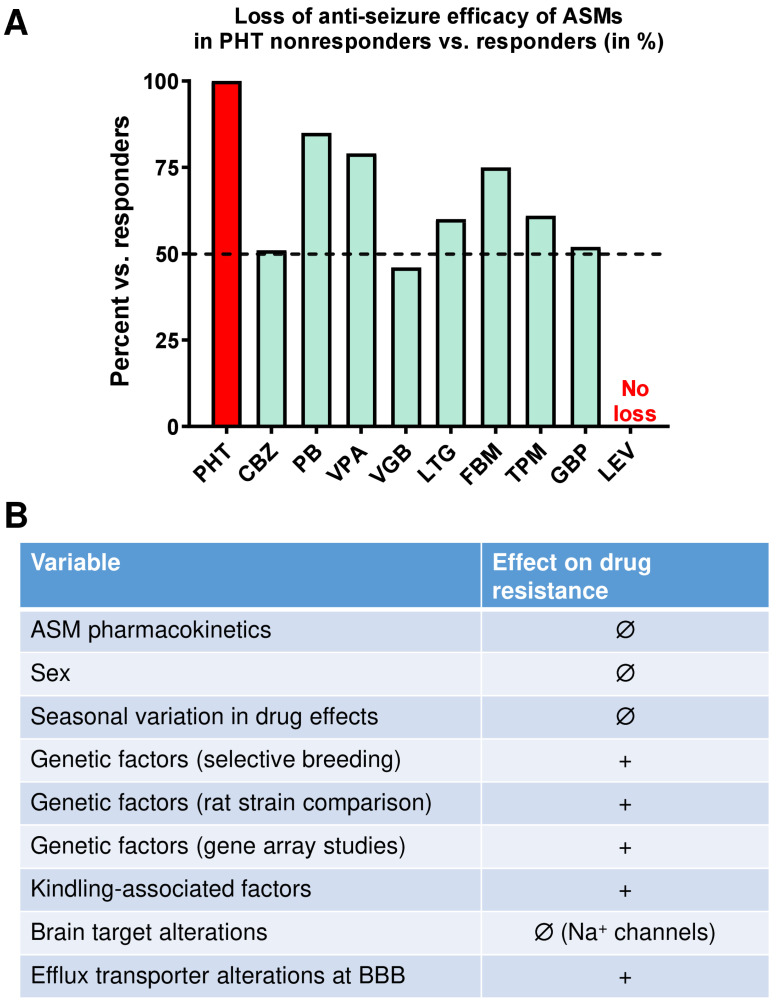
Extension of drug resistance to several anti-seizure medications (ASMs) in the phenytoin-resistant amygdala-kindled rat model and potential mechanisms involved in drug resistance. (**A**) Per definition, the loss of anti-seizure efficacy of phenytoin in phenytoin nonresponders was 100% when compared with the anti-seizure effect of this drug in responders. After kindled rats had been selected into responders and nonresponders by repeated testing with phenytoin or fosphenytoin as shown in Figure 3, each of the other ASMs shown in the figure was tested in at least two different doses in phenytoin responders and nonresponders. The anti-seizure effect was determined by the increase in the focal seizure threshold (ADT) compared with the control ADT in the same rats. Groups of 8–10 fully kindled rats were used for each drug trial. All drugs shown in the figure significantly increased ADT in phenytoin responders. Loss of efficacy in nonresponders is indicated by comparing the drug-induced ADT increase in phenytoin nonresponders with that obtained in responders. Except for levetiracetam, all drugs were less efficacious (by at least 50%) in phenytoin nonresponders compared with responders. Data are from Löscher [31]. (**B**) Various variables were tested for their role in drug resistance in amygdala-kindled phenytoin nonresponders. See text for discussion. Symbols: ∅, no effect; +, effect. Abbreviations: BBB, blood–brain barrier; CBZ, carbamazepine; FBM, felbamate; GBP, gabapentin; LEV, levetiracetam; LTG, lamotrigine; PB, phenobarbital; PHT, phenytoin; TPM, topiramate; VGB, vigabatrin; and VPA, valproate.

**Figure 5 cells-12-01233-f005:**
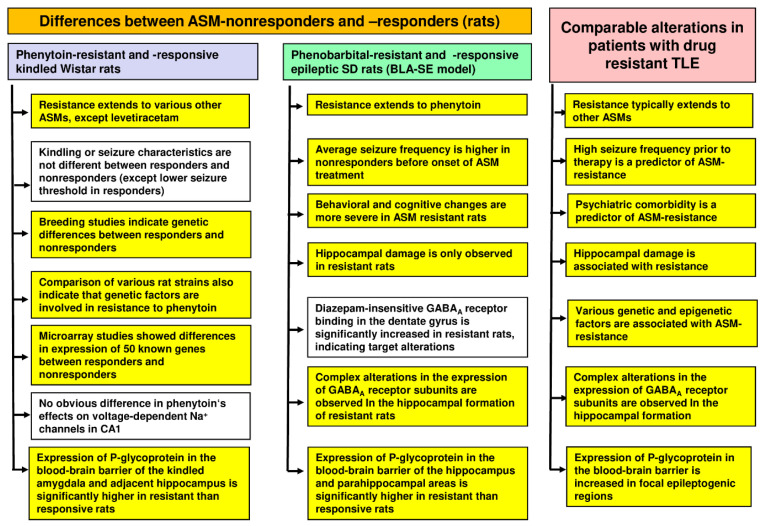
Differences between ASM responders and nonresponders in two animal models of drug-resistant epilepsy. For comparison, alterations associated with ASM resistance in patients are shown. Those alterations that occur both in the models and in patients are highlighted by the colored boxes. For details see Löscher [3], Löscher et al. [15], and Löscher [17].

**Figure 6 cells-12-01233-f006:**
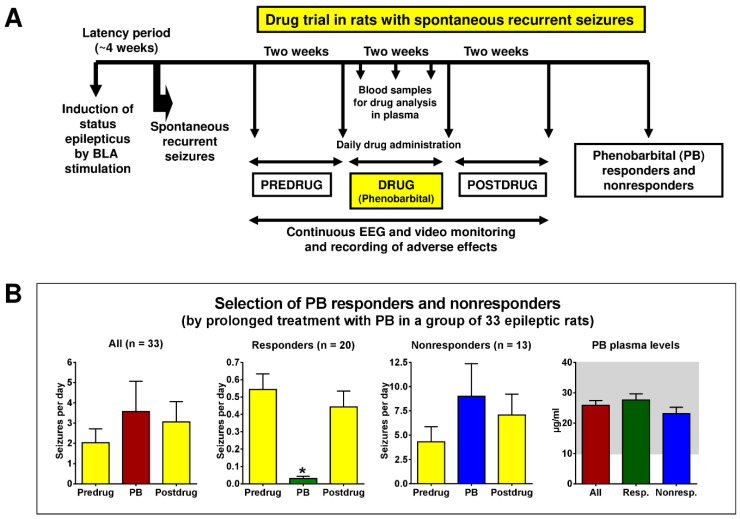
The phenobarbital-resistant epileptic rat. (**A**) Illustrates the procedure to select phenobarbital-responders and nonresponders from large groups of epileptic female Sprague–Dawley rats. Rats were made epileptic in response to SE induced by sustained (~25 min) electrical stimulation of the basolateral amygdala (BLA). The dosing protocol for phenobarbital (PB) consisted of an i.p. bolus dose of 25 mg/kg PB in the morning of the first treatment day, followed 10 h later by an administration of 15 mg/kg i.p., and then twice daily 15 mg/kg i.p. for the 13 subsequent days. This dosing protocol was shown to lead to the maintenance of plasma levels of PB within the therapeutic range (10–40 µg/m). Furthermore, these doses reflected the maximum tolerated doses of PB in rats. (**B**) Illustrates the effect of PB on spontaneous recurrent seizures (SRSs) in 33 epileptic rats from 3 prospective studies (data were combined in this figure). As shown in (**A**), SRSs were recorded over two weeks before the onset of PB treatment (predrug control), followed by drug treatment for two weeks, and then a two-week postdrug control period. SRSs were continuously (24/7) recorded by video EEG in the 33 rats over the 6 weeks of this experiment. A response to PB was defined by complete seizure suppression during treatment or a seizure suppression of >50–75% compared with seizure frequency in the predrug and postdrug control periods. The first graph in (**B**) illustrates individual seizure frequencies (SRSs in 2 weeks) of all 33 rats in these experiments, while the second graph shows respective data from the 20 responders, and the third graph shows data from the 13 nonresponders selected in these experiments from the 33 rats. Only the PB responders exhibited a significant difference in seizure frequency to control recordings (indicated by asterisks; *p* < 0.001), so the response to this ASM was an all-or-none phenomenon. The fourth graph in (**B**) illustrates the average plasma concentration (mean ± SEM) of PB from the blood samples taken during the treatment period. Statistical analysis did not indicate a significant difference in PB plasma levels between groups. The shaded area indicates the therapeutic plasma concentration range of PB in patients with epilepsy, demonstrating that all rats exhibited PB plasma concentrations within this range. Data are from Löscher [17].

**Figure 7 cells-12-01233-f007:**
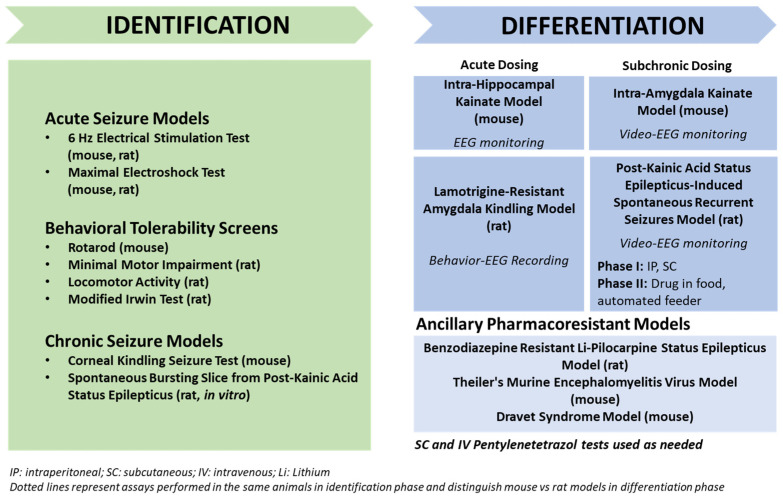
Current testing scheme for the NINDS Epilepsy Therapy Screening Program (ETSP) for pharmacoresistant epilepsy. Models included are those used for the identification and differentiation of potential therapies. Note that for the 6 Hz test, stimulus intensities of 2×C_97_ are used in mice (44 mA) and rats (80 V). Modified from NINDS PANAChE webpage (https://panache.ninds.nih.gov/Home/CurrentModels; assessed on 20 January 2023). See text for details.

**Figure 8 cells-12-01233-f008:**
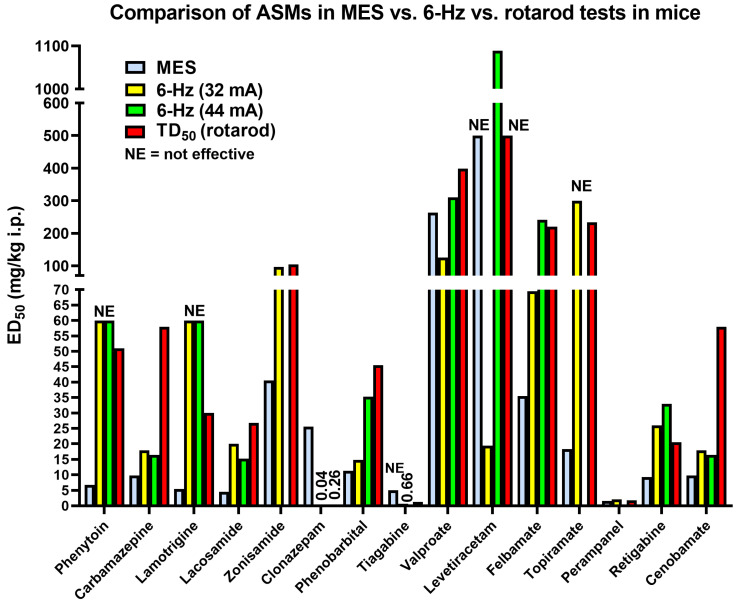
Comparison of anti-seizure potencies of various anti-seizure medications (ASMs) in the maximal electroshock seizure (MES) vs. 6 Hz focal seizure tests in mice. Anti-seizure potencies are illustrated as i.p. ED_50_s. When available, 6 Hz data are shown for 2 corneal stimulation currents, 32 mA and 44 mA. Most data are from male CF-1 mice. For comparison, also i.p. TD_50_s in the rotarod test in mice are shown. “NE” (not effective) indicates that an ED_50_ or TD_50_ could not be determined up to the indicated dose. Note that most ASMs are less effective in the 6 Hz model than in the MES model. Exceptions are clonazepam, tiagabine, valproate, and levetiracetam. Furthermore, most ASMs lose efficacy in the 6 Hz model when increasing the current from 32 to 44 mA. Data are from Barton et al. [113] and Guignet et al. [116]. See also Appendix A for an illustration of these data.

**Figure 9 cells-12-01233-f009:**
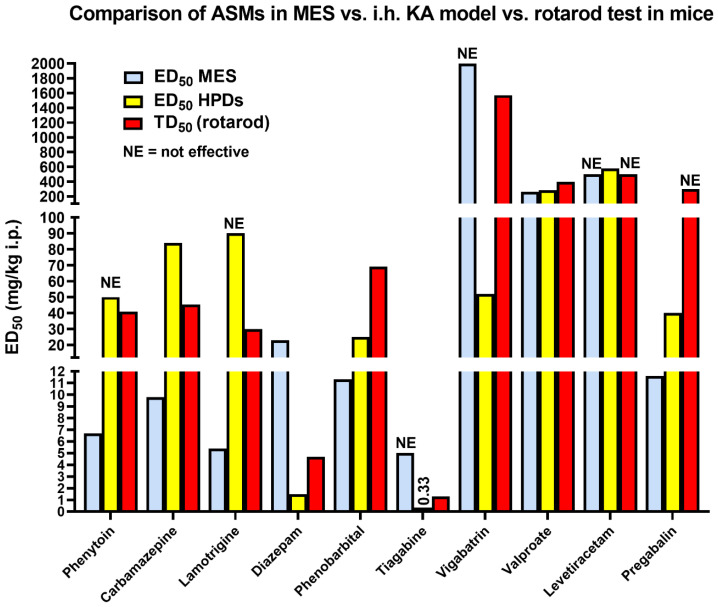
Comparison of anti-seizure potencies of various anti-seizure medications (ASMs) in the maximal electroshock seizure (MES) test vs. the intrahippocampal kainate model in mice. In the intrahippocampal kainate model, ED_50_s (i.p.) were calculated from the number of hippocampal paroxysmal discharges (HPDs) recorded in the EEG for 40 min before and after acute (single drug) drug treatment [165] or 20 min before and 60 min after acute drug treatment [167]. Before treatment, average HPD frequencies ranged between 32 and 46/h [167]. For comparison, TD_50_s determined in the rotarod test in mice are shown [43,116]. “NE” (not effective) indicates that an ED_50_ or TD_50_ could not be determined up to the indicated dose. Note that the 4/10 ASMs tested (phenytoin, carbamazepine, lamotrigine, valproate, and levetiracetam) were either ineffective (NE) or only reduced the frequency of HPDs at doses in the range of the TD_50_, whereas the other 6 drugs (diazepam, phenobarbital, tiagabine, vigabatrin, levetiracetam, and pregabalin) were effective in this model at doses below their TD_50_. Moreover, note that the definition of HPD used by Duveau et al. [167] differs from that used by Riban et al. [165]; see also Twele et al. [165] for defining the different types of electrographic seizures recorded in this model. MES and rotarod data are from Löscher and Nolting [43], Barton et al. [113], and Guignet et al. [116]. See also Appendix A for an illustration of these data.

**Figure 10 cells-12-01233-f010:**
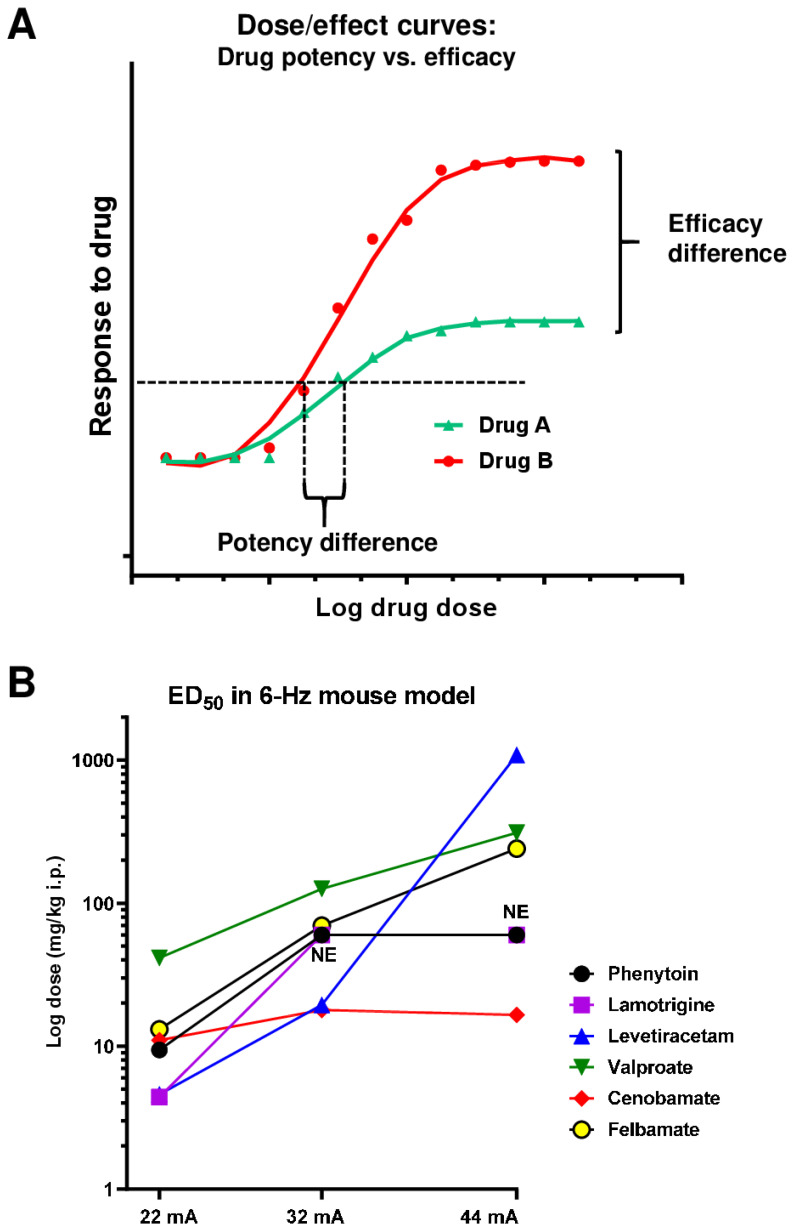
Drug potency vs. efficacy. (**A**) Dose–effect curves of two drugs that illustrate the difference between drug potency and drug efficacy. Drug A has a moderately lower potency (i.e., the dose inducing a defined drug response) but a much lower efficacy (i.e., maximum drug response) than drug B. (**B**) Anti-seizure potency vs. efficacy in the 6 Hz focal seizure mouse model. Anti-seizure potencies are shown as i.p. ED_50_s determined at 3 transcorneal stimulation currents, 22 mA (the CC_97_ for induction of seizures), 32 mA (1.5-times the CC_97_), and 44 mA (twice the CC_97_), respectively. “NE” (not effective) indicates that an ED_50_ could not be determined up to the indicated dose. Note the logarithmic scale used for illustration of ED_50_s. Except for cenobamate, all ASMs shown lose potency with increasing currents, indicating that the anti-seizure efficacy of cenobamate in this model is higher than that of the other ASMs. Data are from Barton et al. [113] and Guignet et al. [116].

**Table 2 cells-12-01233-t002:** Elimination half-life of clinically approved antiseizure medications in adult humans, rats, and mice. Data are from Löscher and Klein [217]. ”?” indicates that no data were found with the Pubmed database.

Medication	Elimination Half-Life (h)			Comments
	Humans	Rats	Mice	
Acetazolamide	10–15	0.33	?	
Brivaracetam	7–8	2.8	?	
Cannabidiol	18–32	7.8	4.7	
Carbamazepine	25–50	1.2–3.5	3.4	Active metabolite = carbamazepine-10,11-epoxide; reduction in half-life during chronic treatment (autoinduction)
Cenobamate	50–60	2.9	?	
Clobazam	10–30	1	0.25	Active metabolite = norclobazam
Clonazepam	17–56	?	2.1	
Diazepam	30–56 (nordazepam 36–200)	0.88 (nordazepam 1.1)	0.67 (nordazepam > 4 h)	Active metabolites = nordazepam (main metabolite), oxazepam, and temazepam
Eslicarbazepine acetate	10–20	?	5.2	Half-lives refer to active metabolite = (*S*)-licarbazepine (eslicarbazepine)
Ethosuximide	40–60	10–16	?	
Everolimus	~30	20	4.3	Long persistence in the brain
Felbamate	16–22	2–17	?	In rodents, non-linear kinetics (half-life increases with increasing doses)
Fenfluramine	13–30	2.6	4.3	Active metabolite = norfenfluramine
Gabapentin	5–9	2–3	?	
Lacosamide	13	3	?	
Lamotrigine	15–35	12–30		
Levetiracetam	6–8	2–3	1.5	
Oxcarbazepine	8–15	0.7–4	6.8	Half-lives refer to active metabolite = (*S*)-licarbazepine (eslicarbazepine)
Perampanel	70	2	?	
Phenobarbital	70–140	9–20	4–7.5	Reduction in half-life during chronic treatment (autoinduction)
Phenytoin	15–20	~2	5–16	Non-linear kinetics (half-life increases with increasing doses); autoinduction
Pregabalin	5–7	~2	~2	
Primidone	6–12	5	2.2	Active metabolite = phenobarbital; autoinduction
Retigabine (ezogabine)	6–8	~2	?	
Rufinamide	6–10	~8	?	
Stiripentol	4.5–13	13	?	
Sulthiame	2–16	?	?	
Tiagabine	5–9	1	?	
Topiramate	20–30	2.5	?	
Valproate	8–15	~1.5	0.8	In rodents, non-linear kinetics (half-life increases with increasing doses)
Vigabatrin	5–8	~1	?	Duration of action independent of half-life because of irreversible inhibition of GABA degradation
Zonisamide	50–70	8	?	

## Data Availability

Not applicable.

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
