# Peer review of "Animal Models of Drug-Resistant Epilepsy as Tools for Deciphering the Cellular and Molecular Mechanisms of Pharmacoresistance and Discovering More Effective Treatments"

_cells, 2023, doi:10.3390/cells12091233_

Round 1
Author Response
We thank the reviewers for their thoughtful and constructive comments, which greatly helped to improve the manuscript. In the following, we describe how we dealt with each of the reviewers’ comments. All changes are highlighted in the revised manuscript in yellow.
#Reviewer 1
This review comprehensively describes animal models of focal epilepsy models of pharmacoresistance. This includes a discussion of selection of responders and nonresponders in an effort to better understand mechanisms of drug-refractory epilepsy (DRE). The review clearly and thoroughly describes the major issues related to animal models of DRE, including the poor understanding of mechanisms of pharmacoresistance.
I have only minor concerns and suggested edits.
- Figure 1 lists different categories of mouse and rat models of drug-resistant seizures. The lower half of this figure includes models that are resistant to some ASMs. This includes the mouse 6 Hz model (32 mA). While the differences between 32 mA and 44 mA are discussed at other parts of the manuscript, it is unclear why 32 mA, and not the more pharmacoresistant intensity 44 mA is listed in this figure.
Response: Thank you for indicating this inconsistency. We now show the 44 mA version of the mouse model (80 V version of the rat model) in Fig. 1.
- Page 7, line 258. The author lists gender with respect to the discussion in sex-based differences and the selection of responders and nonresponders. It is more appropriate to list sex in this context.
Response: We agree and now use “sex” throughout the text.
- The second panel in Figure 3B (yellow/green bar graphs) is missing a legend. This presumably describes saline (yellow) and phenytoin (green) treatment?
Response: Yes. This has been corrected.
- There may be a spelling error on page 10, line 345 “Aresponders”. Please confirm
Response: This has been corrected.
- Figure 7. The figure legend, and possibly the describing text on page 20-21, should note that the NINDS ETSP uses the more pharmacoresistant 44 mA stimulus intensity for the 6 Hz model.
Response: We agree and added this information as suggested.
- Figure 8 lists CD-1 mice in the figure title but the figure legend states that most data come from the CF-1 mouse strain. The figure legend also lists Barton et al. 2000 – this may be intended to say Barton et al. 2001 – please confirm
Response: This has been corrected.

Reviewer 2 Report
The review by Losher and White presents an extensive discussion of animal models of drug resistant epilepsy (DRE) allowing the selection of ASM responders and nonresponders, and underlying mechanisms of pharmacoresistance. The review discusses the difference of DRE investigations on induced and spontaneous seizures in animal models of epilepsy. Finally, it points out the importance to distinguish between drug potency and drug efficacy, and the importance to take into account pharmacokinetic aspects during the experimental design in order to perform a clinical relevant investigation. Below, some comments the authors should address to implement their review.
Major comments:
- In several sentences in the Introduction section, the authors cite reviews or commentary article, most of the time written by themselves, instead of citing the research articles that originally demonstrated the statement. The number of review in the biblio list should be considerable reduced and replaced with original articles. In addition, self-citation is extensive and not justified; authors should reduce them to the strictly necessary.
- Similarly, at page 27, lines 1117-1118, “First studies on the consequences of IHK in rats and mice were published some 40 years ago” please cite original papers not a review from last year.
- Page 26, lines 1099-1102, “...pharmacology of SRS developing in ...traumatic brain injury (TBI) ... has not been characterized to any extent because, in contrast to post-SE models, only a small percentage of rodents develop SRS in such models...”; this statement is not correct as D’Ambrosio’s group demonstrated that within 4 week post severe lateral fluid percussion injury (FPI) more than 80% of rats develop unprovoked seizures (D’Ambrosio et al., 2004). In addition, the authors should include papers by D’Ambrosio’s group (Eastman et al., 2010; Eastman et al., 2011; Eastman et al., 2021) investigating the pharmacoresistance of severe FPI PTE model to carbamazepine and carisbamate, while valproate and brivaracetam showed some positive effects of the disease. Authors should implement this part of the review and complete with the missed studies.
- Page 28, lines 1161-1166: “Compared to models with systemic administration of kainate or pilocarpine, one important advantage of the IHK mouse model is that the neuropathology in this model is much more restricted,...observed in patients with TLE”. In the pilocarpine model, damage is widespread and involves several brain regions, and this is in line with what observed in TLE patients where neuronal lesions have been described in amygdala, thalamus and neocortex, other than hippocampus (Mathern et al., 1997). Therefore it is not correct to state that the IHK model ,characterized by a more restricted hippocampal sclerosis, provides advantages compared to models with systemic administration of kainate or pilocarpine. Authors should fix this error.
Minor comments:
- among different types of DRE available, the authors chose to restrict the review to models of TLE; this choice is reasonable as information on TLE as a DRE is extensive, however, this restriction should be clear from the title. I suggest authors reformulate the title including temporal lobe epilepsy.
- Pag. 1, last line (44), replace “This explains whyalmost all animal models” with “This explains why almost all animal models...”
- Fig. 1 cannot be easily read. Increase font of text.
- Fig. 4, in the legend add symbols used in figure.
- Page 10, line 345, replace “Aresponders and nonresponders” with “responders and nonresponders”.
- Page 11, line 406, BBB abbreviation was already used and explained at page 2, line 81; authors can avoid to repeat the full description here.
Author Response
We thank the reviewers for their thoughtful and constructive comments, which greatly helped to improve the manuscript. In the following, we describe how we dealt with each of the reviewers’ comments. All changes are highlighted in the revised manuscript in yellow.
Reviewer 2
The review by Losher and White presents an extensive discussion of animal models of drug resistant epilepsy (DRE) allowing the selection of ASM responders and nonresponders, and underlying mechanisms of pharmacoresistance. The review discusses the difference of DRE investigations on induced and spontaneous seizures in animal models of epilepsy. Finally, it points out the importance to distinguish between drug potency and drug efficacy, and the importance to take into account pharmacokinetic aspects during the experimental design in order to perform a clinical relevant investigation. Below, some comments the authors should address to implement their review.
Major comments:
- In several sentences in the Introduction section, the authors cite reviews or commentary article, most of the time written by themselves, instead of citing the research articles that originally demonstrated the statement. The number of review in the biblio list should be considerable reduced and replaced with original articles. In addition, self-citation is extensive and not justified; authors should reduce them to the strictly necessary.
Response: W. Löscher’s group discovered pharmacoresistant subgroups of amygdala-kindled Wistar rats in the late 1980s and characterized them in detail over more than a decade before other groups also used this novel model of ASM resistance. Furthermore, Löscher’s group was the first to demonstrate pharmacoresistant subgroups in other epilepsy models. Similarly, the group of Steve White developed several of the models of drug-resistant epilepsy that are widely used today. Thus, a review of animal models of drug-resistant epilepsy should reflect the history of such models and the role of the authors in this respect. However, we have tried to cover also findings from various other groups in the field and cited numerous original articles. Furthermore, during revision, we added 26 papers from other groups. And we added original articles in the Introduction section.
- Similarly, at page 27, lines 1117-1118, “First studies on the consequences of IHK in rats and mice were published some 40 years ago” please cite original papers not a review from last year.
Response: We agree and now cite the original papers.
- Page 26, lines 1099-1102, “...pharmacology of SRS developing in ...traumatic brain injury (TBI) ... has not been characterized to any extent because, in contrast to post-SE models, only a small percentage of rodents develop SRS in such models...”; this statement is not correct as D’Ambrosio’s group demonstrated that within 4 week post severe lateral fluid percussion injury (FPI) more than 80% of rats develop unprovoked seizures (D’Ambrosio et al., 2004). In addition, the authors should include papers by D’Ambrosio’s group (Eastman et al., 2010; Eastman et al., 2011; Eastman et al., 2021) investigating the pharmacoresistance of severe FPI PTE model to carbamazepine and carisbamate, while valproate and brivaracetam showed some positive effects of the disease. Authors should implement this part of the review and complete with the missed studies.
Response: We agree and now describe the findings of Ray D’Ambrosio’s group although these findings, particularly the definition of seizures, have been a matter of debate.
- Page 28, lines 1161-1166: “Compared to models with systemic administration of kainate or pilocarpine, one important advantage of the IHK mouse model is that the neuropathology in this model is much more restricted,...observed in patients with TLE”. In the pilocarpine model, damage is widespread and involves several brain regions, and this is in line with what observed in TLE patients where neuronal lesions have been described in amygdala, thalamus and neocortex, other than hippocampus (Mathern et al., 1997). Therefore it is not correct to state that the IHK model ,characterized by a more restricted hippocampal sclerosis, provides advantages compared to models with systemic administration of kainate or pilocarpine. Authors should fix this error.
Response: We respectfully disagree. Systemic administration of convulsants such as pilocarpine and kainate induces bilateral damage in a variety of brain regions throughout the brain, including many regions that are not affected in TLE patients. This has been one of the main criticisms about such models in the past, both in the preclinical and clinical arena. W. Löscher’s group has used both the systemic and the focal models over decades and characterized and compared both their histopathology and pharmacology. In contrast to systemic administration, focal kainate injection induces mostly unilateral damage that is not restricted to the hippocampus but also affects other regions, including the thalamus but not the many other regions (including the midbrain) affected by systemic administration. We have now included “unilateral” (i.e., ipsilateral) in the description of the neuropathology and revised the respective statement.
Minor comments:
- among different types of DRE available, the authors chose to restrict the review to models of TLE; this choice is reasonable as information on TLE as a DRE is extensive, however, this restriction should be clear from the title. I suggest authors reformulate the title including temporal lobe epilepsy.
Response: We respectfully disagree. The review contains a long section (4.3) on genetic models, in particular on mouse and zebrafish models of Dravet syndrome. Furthermore, we now added a new section (4.2) on models of neocortical epilepsy.
- Pag. 1, last line (44), replace “This explains whyalmost all animal models” with “This explains why almost all animal models...”
Response: Has been corrected.
- Fig. 1 cannot be easily read. Increase font of text.
Response: Done.
- Fig. 4, in the legend add symbols used in figure.
Response: Done.
- Page 10, line 345, replace “Aresponders and nonresponders” with “responders and nonresponders”.
Response: Done.
- Page 11, line 406, BBB abbreviation was already used and explained at page 2, line 81; authors can avoid to repeat the full description here.
Response: Done.

Round 2
Reviewer 2 Report
Modifications made by the authors, and justifications provided, are appreciated and satisfactory.